# Bayesian Transformed Gaussian Processes

**Xinran Zhu**                                                                        *xz584@cornell.edu*
*Cornell University, Center for Applied Mathematics*

**Leo Huang**[*]                                                                      *ah839@cornell.edu*
*Cornell University, Department of Computer Science*

**Eric Hans Lee**[*]                                                                  *eric.lee@intel.com*
*SigOpt: An Intel Company*

**Cameron Ibrahim**                                                                   *cibrahim@udel.edu*
*University of Delaware, Department of Computer and Information Sciences*

**David Bindel**                                                                      *bindel@cornell.edu*
*Cornell University, Department of Computer Science*

**Reviewed on OpenReview:** *https://openreview.net/forum?id=4zCgjqjzAv*

## Abstract

The Bayesian transformed Gaussian (BTG) model, proposed by Kedem and Oliviera in 1997, was developed as a Bayesian approach to trans-Kriging in the spatial statistics community. In this paper, we revisit BTG in the context of modern Gaussian process literature by framing it as a fully Bayesian counterpart to the Warped Gaussian process that marginalizes out a joint prior over input warping and kernel hyperparameters. As with any other fully Bayesian approach, this treatment introduces prohibitively expensive computational overhead; unsurprisingly, the BTG posterior predictive distribution, itself estimated through high-dimensional integration, must be inverted in order to perform model prediction. To address these challenges, we introduce principled numerical techniques for computing with BTG efficiently using a combination of doubly sparse quadrature rules, tight quantile bounds, and rank-one matrix algebra to enable both fast model prediction and model selection. These efficient methods allow us to compute with higher-dimensional datasets and apply BTG with layered transformations that greatly improve its expressibility. We demonstrate that BTG achieves superior empirical performance over MLE-based models in the low-data regime —situations in which MLE tends to overfit.

## 1 Introduction

Gaussian processes (GPs), also known as Kriging models in the spatial statistics community Cressie (1993), are a powerful probabilistic learning framework, including a marginal likelihood which represents the probability of data given only GP hyperparameters. The marginal likelihood automatically balances model fit and complexity terms to favor simple models that explain the data well.

A GP assumes normally distributed observations. In practice, however, this condition is not always adequately met. The classic approach to moderate departures from normality is trans-Gaussian Kriging, which applies a normalizing nonlinear transformation to the data (Cressie, 1993). This idea was reprised and expanded upon in the machine learning literature. One instance is the warped GP (WGP), which maps the observation space to a latent space in which the data is well-modeled by a GP and which learns GP hyperparameters through maximum likelihood estimation (Snelson et al., 2004). The WGP paper employs

---

[*]These authors contributed equally to this work.

a class of parametrized, hyperbolic tangent transformations. Later, Rios & Tobar (2019) introduced compositionally warped GPs (CWGP), which chain together a sequence of parametric transformations with closed form inverses. Bayesian warped GPs further generalize WGPs by modelling the transformation as a GP (Lázaro-Gredilla, 2012). These are in turn generalized to Deep GPs by Damianou & Lawrence (2013), which stack GPs together in the layers of a neural network.

Throughout this line of work, the GP transformation and kernel hyperparameters are typically learned through joint maximum likelihood estimation (MLE). A known drawback of MLE is overconfidence in the data-sparse or low-data regime, which may be exacerbated by warping (Chai & Garnett, 2019). Bayesian approaches, on the other hand, offer a way to account for uncertainty in values of model parameters.

Bayesian trans-Kriging (Spöck et al., 2009) treats both transformation and kernel parameters in a Bayesian fashion. A prototypical Bayesian trans-Kriging model is the BTG model developed by Oliveira et al. (1997). The model places an uninformative prior on the precision hyperparameter and analytically marginalizes it out to obtain a posterior distribution that is a mixture of Student's t-distributions. Then, it uses a numerical integration scheme to marginalize out transformation and remaining kernel parameters. In this latter regard, BTG is consistent with other Bayesian methods in the GP literature, including those of Gibbs (1998); Adams et al. (2009); Lalchand & Rasmussen (2020).

While BTG was shown to have improved prediction accuracy and better uncertainty propagation, it comes with several computational challenges, which hinder its scalability and limit its competitiveness with the MLE approach.

First, the cost of numerical integration in BTG scales with the dimension of hyperparameter space, which can be large when transform and noise model parameters are both incorporated. Traditional methods such as Monte Carlo (MC) suffer from slow convergence. As such, we leverage sparse grid quadrature and quasi Monte Carlo (QMC), which have a higher degree of precision but require a sufficiently smooth integrand. Second, the posterior mean predictor of BTG is not guaranteed to exist, hence the need to use the posterior *median* predictor. The posterior median and credible intervals do not generally have closed forms, so one must resort to expensive numerical root-finding to compute them. Finally, while fast cross-validation schemes are known for vanilla GP models, leave-one-out-cross-validation (LOOCV) on BTG, which incurs quartic cost naively, is less straightforward to perform because of an embedded generalized least squares problem.

In this paper, we reduce the overall computational cost of end-to-end BTG inference, including model prediction and selection. Our main contributions follow.

- We revisit the Bayesian transformed Gaussian (BTG) model in the context of machine learning literature and GPs. We show that BTG fills an important gap in the GP research in which a fully Bayesian treatment of input-warped GPs has not been thoroughly investigated.

- We extend BTG to support *multi-layered* transformations.

- We propose efficient methods for computing BTG predictive medians and predictive quantiles through a combination of doubly sparse quadrature and quantile bounds. We also propose fast LOOCV using rank-one matrix algebra.

- We develop a framework to control the tradeoff between speed and accuracy for BTG and analyze the error in sparsifying QMC and sparse grid quadrature rules.

- We conduct comparisons between the Bayesian and MLE approaches and provide experimental results for BTG and WGP coupled with 1-layer and 2-layer transformations. We find evidence that BTG is well-suited for low-data regimes, where hyperparameters are under-specified by the data. In these regimes, our empirical testing suggests that BTG provides superior point estimation and uncertainty quantification.

- We develop a modular Julia package for computing with transformed GPs (e.g., BTG and WGP) which exploits vectorized linear algebra operations and supports MLE and Bayesian inference[1].

---

[1]The code used to run experiments in this paper can be found at `https://github.com/xinranzhu/BTG`.

## 2 Background

In this section, we provide a brief overview of Gaussian processes and their transformed counterparts. This section is by no means comprehensive, and is meant to establish common notation and definitions for the reader's convenience. Note that we modified the standard GP treatment of Rasmussen & Williams (2008) to improve clarity when explaining the BTG model; notational differences aside, our formulations are equivalent.

### 2.1 Gaussian Process Regression

A GP $f \sim \mathcal{GP}(\mu, \tau^{-1}k)$ is a distribution over functions in $\mathbb{R}^d$, where $\mu(\boldsymbol{x})$ is the expected value of $f(\boldsymbol{x})$ and $\tau^{-1}k(\boldsymbol{x}, \boldsymbol{x}')$ is the positive (semi)-definite covariance between $f(\boldsymbol{x})$ and $f(\boldsymbol{x}')$. For later clarity, we separate the precision hyperparameter $\tau$ from lengthscales and other kernel hyperparameters (typically denoted by $\theta$) [2]. Unless otherwise specified, we assume a linear mean field and the squared exponential kernel:

$$\mu_\beta(\boldsymbol{x}) = \beta^T m(\boldsymbol{x}), \qquad m \colon \mathbb{R}^d \to \mathbb{R}^p,$$

$$k_\theta(\boldsymbol{x}, \mathbf{x}') = \exp\Big( -\frac{1}{2}\|\boldsymbol{x} - \mathbf{x}'\|_{\mathbf{D}_\theta^{-2}}^2 \Big).$$

Here $m(\boldsymbol{x})$ is a known function mapping a location $\boldsymbol{x}$ to a vector of covariates, $\beta$ is a vector of coefficients, and $D_\theta^2$ is a diagonal matrix of length scales determined by the parameter(s) $\theta$.

For any finite set of input locations, let:

$$X = [\boldsymbol{x}_1, \ldots, \boldsymbol{x}_n]^T \qquad\qquad X \in \mathbb{R}^{n \times d},$$
$$M_X = [m(\boldsymbol{x}_1), \ldots, m(\boldsymbol{x}_n)]^T \qquad\qquad M_X \in \mathbb{R}^{n \times p},$$
$$f_X = [f(\boldsymbol{x}_1), \ldots, f(\boldsymbol{x}_n)]^T \qquad\qquad f_X \in \mathbb{R}^n,$$

where $X$ is the matrix of observations locations, $M_X$ is the matrix of covariates at $X$, and $f_X$ is the vector of observations. A GP has the property that any finite number of evaluations of $f$ will have a joint Gaussian distribution: $f_X \mid \beta, \tau, \theta \sim \mathcal{N}(M_X\beta, \tau^{-1}K_X)$, where $(\tau^{-1}K_X)_{ij} = \tau^{-1}k_\theta(\boldsymbol{x}_i, \boldsymbol{x}_j)$ is the covariance matrix of $f_X$. We assume $M_X$ to be full rank.

Typically, the kernel hyperparameters $\tau$, $\beta$, and $\theta$ are fit by minimizing the negative log likelihood:

$$-\log \mathcal{L}(f_X \mid X, \beta, \tau, \theta) \propto$$
$$\frac{1}{2}\big\|f_X - M_X\beta\big\|_{K_X^{-1}}^2 + \frac{1}{2}\log|K_X|.$$

This is known as type II maximum likelihood estimation (MLE) of the kernel hyperparameters. Having performed MLE to compute the optimal $\beta$, $\tau$, and $\theta$, the posterior predictive density of a point $\boldsymbol{x}$ is:

**Definition 1** (*Posterior predictive density, GP model, type II MLE*)**.**

$$f(\boldsymbol{x}) \mid \beta, \ \tau, \ \theta, \ f_X, X \sim \mathcal{N}(\mu_{\theta,\beta}, s_{\theta,\beta}).$$

$\mathcal{N}(\mu_{\theta,\beta}, s_{\theta,\beta})$ is a normal distribution with the following mean and variance:

$$\mu_{\theta,\beta} = \beta^T m(\boldsymbol{x}) + K_{X\boldsymbol{x}}^T K_X^{-1}(f_X - M_X\beta),$$
$$s_{\theta,\beta} = \tau^{-1}\big(k_\theta(x, x) - K_{X\boldsymbol{x}}^T K_X^{-1} K_{X\boldsymbol{x}}\big),$$

where $(K_{X\boldsymbol{x}})_i = k_\theta(\boldsymbol{x}_i, \boldsymbol{x})$. The Bayesian model selection approach forgoes MLE of the negative log likelihood, and instead assumes a prior distribution over kernel hyperparameters $p(\beta, \tau, \theta)$. This induces a posterior distribution over the kernel hyperparameters by noting that the posterior is proportional to the likelihood times the prior:

---

[2]Standard GP literature includes an additional hyperparameter $\sigma$ which models the observed noise. We omit $\sigma$ in this section for brevity but treat it in our experiments.

$$p(\beta, \tau, \theta \mid f_X, X) \propto \mathcal{L}(f_X \mid X, \beta, \tau, \theta) p(\beta, \tau, \theta).$$

Given this posterior distribution, the posterior predictive density is given by the following.

**Definition 2** (*Posterior predictive density, fully Bayesian GP*)**.**

$$p(f(\boldsymbol{x}) \mid f_X, X) = \int_{\beta, \tau, \theta} p(f(\boldsymbol{x}) \mid \beta, \ \tau, \ \theta, \ f_X, X) \, p(\beta, \tau, \theta \mid f_X, X).$$

We call this the *fully Bayesian* approach to Gaussian process modeling. The well-known tradeoff between type II MLE and the fully Bayesian approach is that of tractability; the fully Bayesian approach requires computing integrals with no generic, analytic solution (often through MCMC) where as type II MLE is much more straightforward.

## 2.2 Warped Gaussian Processes

While GPs are powerful tools for modeling nonlinear functions, they make the fairly strong assumptions of Gaussianity and homoscedasticity. WGPs (Snelson et al., 2004) challenge these assumptions by transforming ("warping") the observation space to a latent space, which itself is modeled by a GP. Given a strictly increasing, differentiable parametric transformation $g_\lambda$ determined by a parameter $\lambda$, WGPs model the composite function $g_\lambda \circ f$ with a GP.

**Definition 3** (*Warped Gaussian Process*)**.**

$$(g_\lambda \circ f) \mid \beta, \tau, \lambda, \theta \sim \mathcal{GP}(\mu_\beta, \tau^{-1} k_\theta).$$

Let $(g_\lambda(f_X))_i = g_\lambda(f(\boldsymbol{x}_i))$. WGP jointly computes the parameters through MLE in the latent space, where the negative log likelihood is:

$$-\log \mathcal{L}\big(g_\lambda(f_X) \mid X, \beta, \tau, \theta, \lambda\big) \propto$$
$$\frac{1}{2}\big\|g_\lambda(f_X) - M_X \beta\big\|^2_{K_X^{-1}} + \frac{1}{2}\log|K_X| - \log J_\lambda.$$

Note that the difference between this equation and the negative log likelihood is the inclusion of $J$, which represents the Jacobian of the transformation:

$$J_\lambda = \left| \prod_{i=1}^n \frac{\partial}{\partial f(\boldsymbol{x}_i)} g_\lambda(f(\boldsymbol{x}_i)) \right|.$$

WGPs predict the value of a point $x$ by computing its posterior mean in the latent space and then inverting the transformation back to the observation space: $g_\lambda^{-1}(\hat{\mu}(x))$. Snelson et al. (2004) uses the tanh transform family, whose members do not generally have closed form inverses; they must be computed numerically.

## 2.3 Other Transformed GP Models

There exist a variety of other transformed GP models, and in this section we briefly summarize a few of them. This overview is by no means comprehensive, but we hope it succinctly communicates not only the breadth of existing work, but also the gaps that we believe BTG fills.

The compositionally warped GP (CWGP) (Rios & Tobar, 2019) considers a set of augmented warping functions used by the standard WGP, which possess analytic inverses (for computational simplicity) and are then composed together for additional expressiveness.

The Bayesian warped GP (BWGP) (Lázaro-Gredilla, 2012) takes a different approach to transformations, placing a GP prior on the warping function and variationally marginalizing it out so that model prediction and selection does not require numerical integration.

Table 1: A table summarizing some of the different GP models in the literature today. The main differences among models are whether or not they use transformations (all do except for the standard GP), what sorts of transformations are used, and what type of inference method is used. *Single* denotes a single function used for transformations. *Multiple* refers to a composition of such functions. At the bottom of the table is the topic of this paper, the Bayesian transformed GP (BTG).

| Name | | Transformations | Type | Inference |
|---|---|---|---|---|
| GP-MLE | | No | N/A | MLE |
| GP-Bayesian | | No | N/A | Fully Bayesian |
| Warped GP | (WGP) | Yes | Single Analytic | MLE |
| Compositionally Warped GP | (CWGP) | Yes | Multiple Analytic | MLE |
| Bayesian Warped GP | (BWGP) | Yes | Single GP | Variational |
| Deep GP | (DGP) | Yes | Multiple GPs | Variational |
| Bayesian Transformed GP | (BTG) | Yes | Multiple Analytic | Fully Bayesian |

Deep GPs (Damianou & Lawrence, 2013) extends the "GP of a GP" idea by building a network of GPs, and also uses variational inference to accelerate model prediction and selection by avoiding numerical integration. Indeed, a whole body of literature exists solely towards the application of variational inference to Gaussian process regression (Tran et al., 2015; Salimbeni & Deisenroth, 2017; Jakkala, 2021).

Like all these other models, BTG considers a family of warping functions —single layer or multiple compositions. However, instead of using type II MLE (which can sometimes be overconfident) or variational inference (VI) (whose approximations often require significant data before showing their advantages), BTG opts for a carefully chosen prior and explicit numerical integration during the model selection process for a fully Bayesian approach to transformed Gaussian process regression.

While we focus on improving the Bayesian treatment and scale BTG to higher dimensions using exact kernel inference, another line of work, orthogonal to ours, focuses on combining the warping setting with sparse GP techniques to scale to large datasets. Graßhoff et al. (2020) adopt the Structured Kernel Interpolation (SKI) method (Wilson & Nickisch, 2015) to the WGP setting, which scales WGP up to large training data in low dimensions. Rossi et al. (2021) considers the sparse GP setting with inducing points and focuses on a fully Bayesian treatment of both the inducing points and model hyperparameters, based on stochastic gradient Hamiltonian Monte Carlo.

## 3  The Bayesian Transformed Gaussian Process (BTG) Model

One might think of the Bayesian Transformed Gaussian (BTG) model (Oliveira et al., 1997) as a fully Bayesian generalization of WGP. WGP uses MLE to learn optimal values for the transformation parameters $\lambda$, mean function parameters $\beta$, signal variance parameters $\tau$, and GP lengthscales $\theta$.

Just like WGP, BTG models a function $f(\boldsymbol{x})$ as:

$$(g_\lambda \circ f) \mid \beta, \tau, \lambda, \theta \sim \mathcal{GP}(\mu_\beta, \tau^{-1} k_\theta).$$

Unlike WGP, BTG places a joint prior $p(\beta, \tau, \theta, \lambda)$ over *all* model parameters and performs fully Bayesian model selection by computing the usual marginal likelihood instead of performing the MLE. A key idea of BTG is to carefully select a joint prior over $\lambda$, $\beta$, $\tau$, and $\theta$ to simplify the computation so that the BTG's posterior predictive density is a mixture of t-distributions —this will be derived in later subsections.

**Definition 4** (Posterior predictive density, BTG)**.**

$$f(\boldsymbol{x}) \mid f_X, X \sim \sum_{i=1}^{M} w_i T_i^{n-p}$$

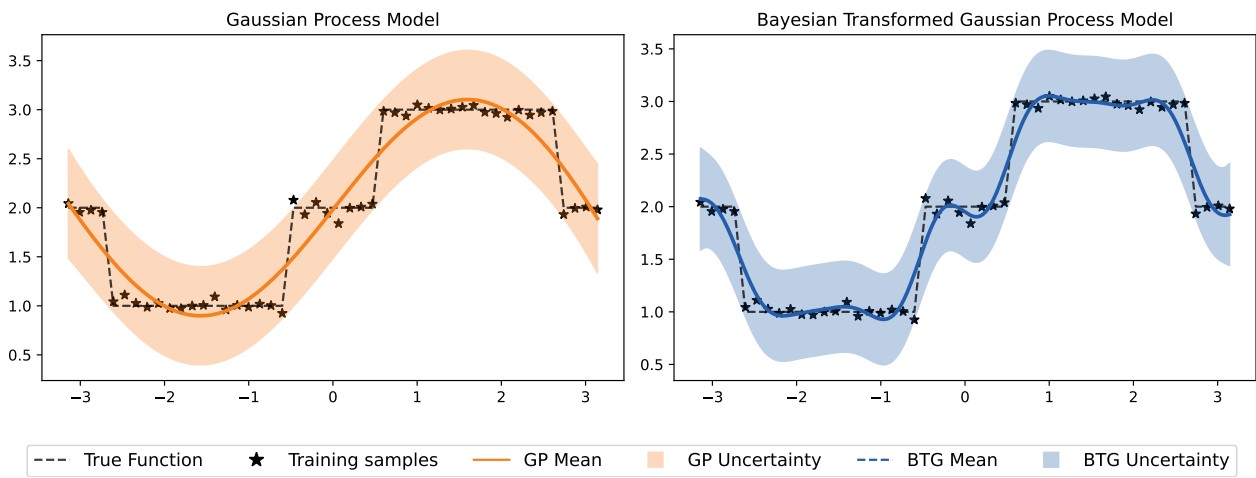

Figure 1: A comparison of GP and BTG, both trained on 48 random samples from the rounded sine function with noise from $\mathcal{N}(0, 0.05)$. Left: the predictive mean and 95% equal tailed credible interval of a Gaussian process using the squared exponential kernel. Right: the predictive median and 95% equal tailed credible interval of BTG using the squared exponential kernel. BTG is better able to capture the irregularity in the training data despite a highly smooth kernel, largely due to it's Bayesian treatment of warping functions that map the data points to a latent space.

where each $T_i^{n-p}$ is a different transformed $T$ distribution whose mean and variance we will derive in §3.2 and §4; for a comprehensive analysis, see Box & Cox (1964). Like in the case of the WGP, this predictive distribution must then be inverted to perform prediction or uncertainty quantification.

BTG appears to possess the same advantages that any fully Bayesian model will possess over it's MLE counterpart: robustness to overconfidence. This results in a better fit, as demonstrated in Figure 1. The underlying kernel is the highly smooth squared exponential, resulting in a GP that is unable to capture the underlying function, which is not smooth at all. Due to a combination of input warping and Bayesian inference, BTG is able to resolve the underlying datapoints much better than a GP despite using the same smooth kernel. In later sections, we explore the advantages of being Bayesian in the low-data regime.

### 3.1 The Parameter Posterior

A key idea of the BTG model is that, conditioned on $\lambda$, $\theta$, and $f_X$, the resulting WGP is a generalized linear model (Oliveira et al., 1997). We estimate $\beta$ by $\hat{\beta}_{\theta,\lambda}$, the solution to the weighted least squares problem:

$$q_{\theta,\lambda} = \min_{\beta} \left\| g_\lambda(f_X) - M_X \beta \right\|_{K_X^{-1}}^2,$$

where $q_{\theta,\lambda}$ is the residual norm. Explicity, we have

$$\hat{\beta}_{\theta,\lambda} = (M_X^T K_X^{-1} M_X)^{-1} M_X^T K_X^{-1} g_\lambda(f_X),$$
$$q_{\theta,\lambda} = (g_\lambda(f_X) - M_X \hat{\beta}_{\theta,\lambda})^T K_X^{-1} (g_\lambda(f_X) - M_X \hat{\beta}_{\theta,\lambda}).$$

Therefore, we have the marginal probability

$$p(f_X \mid X) = \int p(f_X | \beta, \tau, \theta, \lambda, X) p(\beta, \tau, \theta, \lambda) \, d\lambda \, d\beta \, d\theta \, d\tau$$
$$= \int |K_X|^{-1/2} |M_X^T K_X^{-1} M_X|^{-1/2} q_{\theta,\lambda}^{-(n-p)/2} J_\lambda^{1-p/n} p(\theta) p(\lambda) \, d\theta \, d\lambda.$$

Further using Bayes theorem, we have

$$p(\theta, \lambda \mid f_X, X) \propto |K_X|^{-1/2} |M_X^T K_X^{-1} M_X|^{-1/2} q_{\theta,\lambda}^{-(n-p)/2} J_\lambda^{1-p/n} p(\theta) p(\lambda). \tag{1}$$

On the other hand, because appropriate values for $\beta$, $\tau$, and $\theta$ depend nontrivially on $\lambda$, BTG adopts the improper joint prior:

$$p(\beta, \tau, \theta, \lambda) \propto p(\theta)p(\lambda) \, / \, (\tau J_\lambda^{p/n}).$$

This effectively decouples the priors on $\theta$ and $\lambda$; $p(\theta)$ and $p(\lambda)$ are independent and user-specified. BTG then selects the prior on $(\beta, \tau)$ to follow the conditional normal-inverse-gamma distribution:

$$\begin{aligned}
\beta \mid \tau, \lambda, \theta, f_X, X &\sim \mathcal{N}\big(\hat{\beta}_{\lambda,\theta}, \tau^{-1}(M_X^T K_X^{-1} M_X)^{-1}\big), \\
\tau \mid \lambda, \theta, f_X, X &\sim \Gamma^{-1}\Big(\frac{n-p}{2}, \frac{2}{q_{\lambda,\theta}}\Big).
\end{aligned} \tag{2}$$

Note that, due to the improper joint prior, the distribution on $\beta$ and $\tau$ depends on the values of $\theta$ and $\lambda$. Using (1) and (2), the posterior distribution, up to some constant, is determined:

$$p(\beta, \tau, \theta, \lambda \mid f_X, X) = p(\beta, \tau \mid \theta, \lambda, f_X, X)p(\theta, \lambda \mid f_X, X). \tag{3}$$

The reason for the normal-inverse-gamma prior becomes apparent; the marginal of a normal-inverse-gamma is a t-distribution, and so once we marginalize out equation 3, the mixture of t-distributions seen in equation 4 will be the result. We derive this in the following subsection.

### 3.2 The Predictive Density

BTG bases the prediction at $\boldsymbol{x}$ on the *Bayesian predictive density function* (Aitchinson & Dunsmore, 1975):

$$p(f(\boldsymbol{x}) \mid f_X, X) = \int p(f(\boldsymbol{x}) \mid \beta, \tau, \lambda, \theta, f_X, X)p(\beta, \tau, \lambda, \theta \mid f_X, X) \, d\lambda \, d\beta \, d\theta \, d\tau. \tag{4}$$

The Bayesian predictive density function is simply an integral of a product of two probability densities. In the previous section, we derived $p(\beta, \tau, \lambda, \theta \mid f_X, X)$, so we know the second probability density. The first probability density is that of a warped GP (Definition 3):

$$\begin{aligned}
g_\lambda \circ f(\boldsymbol{x}) \mid \beta, \tau, \lambda, \theta, f_X, X &\sim \mathcal{N}(m_{\beta,\theta,\lambda}, D_\theta), \\
m_{\beta,\theta,\lambda} &= \beta^T m(\boldsymbol{x}) + K_{X\boldsymbol{x}}^T K_X^{-1}(g_\lambda \circ f_X - M_X\beta), \\
D_\theta &= \tau^{-1}\big(k_\theta(x,x) - K_{X\boldsymbol{x}}^T K_X^{-1} K_{X\boldsymbol{x}}\big).
\end{aligned}$$

We consequently know everything needed to specify Equation 4. Because we adopted a conditional normal-inverse-gamma distribution, once we marginalize out $\beta$ and $\tau$, the posterior of $g_\lambda \circ f(\boldsymbol{x})$ is the *t*-distribution:

$$g_\lambda \circ f(\boldsymbol{x}) \mid \lambda, \theta, f_X, X \sim T^{n-p}\big(m_{\lambda,\theta}, (q_{\theta,\lambda} C_{\theta,\lambda})^{-1}\big), \tag{5}$$

where the mean largely resembles that of a GP:

$$m_{\lambda,\theta} = K_{\boldsymbol{x}X} K_X^{-1}\big(g_\lambda(f_X) - M_X\hat{\beta}_{\lambda,\theta}\big) + \hat{\beta}_{\lambda,\theta}^T m(\boldsymbol{x}),$$

and $C_{\lambda,\theta}$ is the final Schur complement $B(\boldsymbol{x})/[k_\theta(\boldsymbol{x}, \boldsymbol{x})]$ of the bordered matrix:

$$B(\boldsymbol{x}) = \begin{bmatrix} 0 & M_X^T & m(\boldsymbol{x}) \\ M_X & K_X & K_{X\boldsymbol{x}} \\ m(\boldsymbol{x})^T & K_{X\boldsymbol{x}}^T & k_\theta(\boldsymbol{x}, \boldsymbol{x}) \end{bmatrix}.$$

Therefore, by Bayes theorem, the marginal posterior of BTG is:

$$p(f(\boldsymbol{x}) \mid f_X, X) = \frac{\int_{\Theta,\Lambda} \phi(f(\boldsymbol{x}) \mid \theta, \lambda, f_X, X)p(f_X \mid \lambda, \theta)p(\theta)p(\lambda)d\lambda d\theta}{\int_{\Theta,\Lambda} p(\theta, \lambda \mid f_X, X)p(\theta)p(\lambda) \, d\lambda \, d\theta}, \tag{6}$$

where the posterior being integrated is a transformed $t$-distribution with the pdf

$$
\phi(f(\boldsymbol{x}) \mid \theta, \lambda, f_X, X) =
$$
$$
\frac{\Gamma(\frac{n-p+1}{2})|g'_\lambda(f(\boldsymbol{x}))|}{\Gamma(\frac{n-p}{2})\pi^{1/2}|q_{\theta,\lambda}C_{\theta,\lambda}|^{1/2}}[1 + (f(\boldsymbol{x}) - m_{\theta,\lambda})^T(q_{\theta,\lambda}C_{\theta,\lambda})^{-1}(f(\boldsymbol{x}) - m_{\theta,\lambda})]^{-(n-p+1)/2}. \tag{7}
$$

Unlike WGP, BTG may not have first or second moments, because its marginal posterior may be for example, a mixture of log-t distributions. If this occurs, the probability density function (pdf) will not have a mean or variance. Therefore, BTG instead uses the median and credible intervals, computed by inverting its cumulative distribution function (cdf).

### 3.3 The Approximate Predictive Density

For general nonlinear transformations, the posterior distribution of BTG (Equation 6) is intractable and therefore we approximate it using a set of $M$ quadrature nodes and weights $([\theta_i, \lambda_i], w_i)$, yielding the mixture of transformed t-distributions

$$
p\big(f(\mathbf{x}) \mid f_X, X\big) \approx \frac{\sum_{i=1}^{M} w_i\phi\big(f(\boldsymbol{x}) \mid \theta_i, \lambda_i, f_X, X\big)p(f_X|\theta_i,\lambda_i)p(\theta_i)p(\lambda_i)}{\sum_{i=1}^{M} w_i p(\theta_i, \lambda_i \mid f_X)p(\theta_i)p(\lambda_i)}.
$$

Here, $\phi\left(f(\boldsymbol{x}) \mid \theta_i, \lambda_i, f_X, X\right)$ is the pdf of the transformed $t$-distributions (Equation 3.2), $p(f_X|\theta_i,\lambda_i)$ is the likelihood of data given hyperparameters, $p(\theta_i)$ and $p(\lambda_i)$ are the user-specified hyperparameter priors, and $w_i$ is a quadrature weight at the quadrature point $(\theta_i, \lambda_i)$.

To simplify notation, we combine all terms except for the transformed t-distribution pdf into weights $\tilde{w}_i$ to simplify the predictive distribution into our desired mixture of transformed t-distributions:

$$
\begin{aligned}
\tilde{w}_i &:= \frac{w_i p(f_X|\theta_i,\lambda_i)p(\theta_i)p(\lambda_i)}{\sum_{i=1}^{M} w_i p(\theta_i, \lambda_i \mid f_X)p(\theta_i)p(\lambda_i)}, \\
p\big(f(\boldsymbol{x}) \mid f_X, X\big) &\approx \sum_{i=1}^{M} \tilde{w}_i\phi\big(f(\boldsymbol{x}) \mid \theta_i, \lambda_i, f_X, X\big).
\end{aligned} \tag{8}
$$

As mentioned earlier, $p\big(f(\boldsymbol{x}) \mid f_X, X\big)$ is not guaranteed to have a mean, so we must use the median predictor instead. We do so by computing the quantile $P^{-1}(0.5)$ by numerical root-finding, where $P$ is the cdf of $p\big(f(\boldsymbol{x}) \mid f_X, X\big)$, and therefore a mixture of transformed t-distribution cdfs.

## 4 Methodology

BTG regression via the median predictor (or any other quantile) of Equation 8 is challenging. The dimensionality of the integral scales with the total number of hyperparameters, which contains the kernel lengthscale hyperparameters $\theta$ that grow with the ambient dimension of the data, as well as the transformation parameters $\lambda$ that grow with the number of transformations used. Furthermore, its cdf must be numerically inverted, requiring many such quadrature computations for even a single, point-wise regression task. This is further complicated by the difficulty in assessing model fit through LOOCV, which must be repeated at every quadrature node as well. As a result, a naive implementation of BTG scales poorly.

In this section, we discuss and propose efficient algorithms that make BTG model prediction and model validation far faster, and indeed, comparable to the speed of its MLE counterparts. First, we discuss our doubly sparse quadrature rules for computing the BTG predictive distribution (§4.1 and §4.2). We then provide quantile bounds that accelerate the root-finding convergence (§4.3). Next, we propose an $\mathcal{O}(n^3)$ LOOCV algorithm for BTG using Cholesky downdates and rank-1 matrix algebra (§4.4), which greatly improves the naive $\mathcal{O}(n^4)$ LOOCV cost. Finally we discuss the single and multi-layer nonlinear transformations used in our experiments (§4.5).

## 4.1 Sparse Grid Quadrature

Sparse grid methods, or Smolyak algorithms, are effective for approximating integrals of sufficient regularity in moderate to high dimensions. While the conventional Monte Carlo (MC) quadrature approach used by Oliveira et al. (1997) converges at the rate of $\mathcal{O}(1/\sqrt{M})$, where $M$ is the number of quadrature nodes, the approximation error of Smolyak's quadrature rule is $\mathcal{O}\left(M^{-r}|\log_2 M|^{(d-1)(r+1)}\right)$ where $d$ is the dimensionality of the integral and $r$ is the integrand's regularity i.e., number of derivatives.

In this paper, we use a sparse grid rule detailed by Bungartz & Griebel (2004) and used for likelihood approximation by Heiss & Winschel (2008).

## 4.2 Quadrature Sparsification

Numerical integration schemes such as sparse grids and QMC use $M$ fixed quadrature nodes, where $M$ depends on the dimensionality of the domain and fineness of the grid. In the Bayesian approach, expensive GP operations such as computing a log determinant and solving a linear system are repeated across quadrature nodes, for a total time complexity of $\mathcal{O}(Mn^3)$.

In practice, many nodes are associated with negligible mixture weights, so their contribution to the posterior predictive distribution can effectively be ignored. We thus adaptively drop nodes when their associated weights fall below a certain threshold. To do so in a principled way, we approximate the mixture with a subset of dominant weights and then quantify the error in terms of the total mass of discarded weights.

We assume the posterior cdf is the mixture of cdfs $F(x) = \sum_{i=1}^{M} w_i f_i(x)$, where each $f_i$ is a cdf. Assume the weights $\{w_i\}_{i=1}^{M}$ are ordered by decreasing magnitude. Consider $F_k(x)$, a truncated and re-scaled $F(x)$. We first quantify the pointwise approximation error in Lemma 4.1. We then quantify the error in quantile computation: Propositions 4.1, 4.2 show that the approximated quantile $F_k^{-1}(p)$ can be bounded by perturbed true quantiles. Proposition 4.3 gives a simple bound between $F_k^{-1}(p)$ and $F^{-1}(p)$ within the region of interest, and applies to both QMC—which uses positive weights—and sparse grid quadrature—which uses positive *and* negative weights. See Appendix A.1 for proofs.

**Lemma 4.1.** *Let $k$ be the smallest integer such that $\sum_{i=1}^{k} w_i \geq 1 - \epsilon$. Then define the scaled, truncated mixture*

$$F_k(x) := \frac{1}{c} \sum_{i=1}^{k} w_i f_i(x), \quad c := \sum_{i=1}^{k} w_i.$$

*We have*

$$|F(x) - F_k(x)| \leq 2\epsilon.$$

**Proposition 4.1** (Error Bound for Positive Weights)**.** *For any $\epsilon \in (0,1)$, let $k$ be the smallest integer such that $\sum_{i=1}^{k} w_i \geq 1 - \epsilon$. Define the scaled, truncated mixture*

$$F_k(x) := \frac{1}{c} \sum_{i=1}^{k} w_i f_i(x), \quad c := \sum_{i=1}^{k} w_i.$$

*Let $p \in (0,1)$ and assume that $p \pm 2\epsilon \in (0,1)$. Then the approximate quantile $F_k^{-1}(p)$ is bounded by perturbed true quantiles:*

$$F^{-1}(p - 2\epsilon) \leq F_k^{-1}(p) \leq F^{-1}(p + 2\epsilon).$$

**Proposition 4.2** (Error Bound for Negative Weights)**.** *Let $F(x)$ be defined as before, except each $w_i$ is no longer required to be positive. Consider the split $F(x) = F_{M'}(x) + R_{M'}(x)$, where $F_{M'}(x) = \sum_{i=1}^{M'} w_i f_i(x)$ and $R_{M'}(x) = \sum_{i=M'+1}^{M} w_i f_i(x)$. Then for any $x$, we have $R_{M'}(x) \in [\epsilon_-, \epsilon_+]$, where the epsilons are defined as the sum of positive (resp. negative) weights of $R_{M'}(x)$*

$$\epsilon_- = \sum_{i=M'+1}^{M} [w_i]_- \leq 0 \, , \; \epsilon_+ = \sum_{i=M'+1}^{M} [w_i]_+ \geq 0.$$

*Let $p \in (0,1)$ and assume $p + \epsilon_-, p + \epsilon_+ \in (0,1)$. Then the approximate quantile $F_{M'}^{-1}(p)$ is bounded by perturbed true quantiles:*

$$F^{-1}(p + \epsilon_-) \leq F_{M'}^{-1}(p) \leq F^{-1}(p + \epsilon_+).$$

**Proposition 4.3** (Error Bound at a quantile). *Let $F(x)$ be defined as before, $\epsilon_1, \epsilon_2 \in (0,1)$, and $F_k(x)$ be an approximate to $F(x)$ such that $F^{-1}(p - \epsilon_1) \leq F_k^{-1}(p) \leq F^{-1}(p - \epsilon_2)$ for some $p \in (0,1)$. Assuming $p - \epsilon_1, p + \epsilon_2 \in (0,1)$, we have the following error bound at a quantile,*

$$\left| F_k^{-1}(p) - F^{-1}(p) \right| \leq \epsilon \max_{\xi \in (p - \epsilon_1, p + \epsilon_2)} \left| \frac{dF^{-1}}{dx}(\xi) \right|,$$

*where $\epsilon = \max\{\epsilon_1, \epsilon_2\}$.*

By adaptively sparsifying our numerical quadrature schemes, we are able to discard a significant portion of summands in the mixture $F(x)$, which in turn, enables significant speedup of BTG model prediction. Empirical results are shown in §5.

### 4.3 Quantile Bounds

To compute posterior quantiles, we apply Brent's algorithm, a standard root-finding algorithm combining the secant and bisection methods, to the cdf defined in Equation 8. Since Brent's algorithm is box-constrained, we use quantile bounds to narrow down the locations of the quantiles for $p \in \{0.025, 0.5, 0.975\}$.

Let $F(x) = \sum_{i=1}^{M} w_i f_i(x)$. Then we have the following bounds for the quantile $F^{-1}(p)$. See Appendix A.2 for proofs.

**Proposition 4.4** (Convex Hull). *Let $F(x)$ be defined as before with $w_i > 0$ and $\sum_{i=1}^{M} w_i = 1$. Then*

$$\min_i f_i^{-1}(p) \leq F^{-1}(p) \leq \max_i f_i^{-1}(p).$$

**Proposition 4.5** (Singular Weight). *Let $F(x)$ be defined as before with $w_i > 0$ and $\sum_{i=1}^{M} w_i = 1$. Let $\overline{w}_i = 1 - w_i$. Then*

$$\max_{p - \overline{w}_i \geq 0} f_i^{-1}(p - \overline{w}_i) \leq F^{-1}(p) \leq \min_{p + \overline{w}_i \leq 1} f_i^{-1}(p + \overline{w}_i).$$

When solving for $y^* = F^{-1}(p)$, we run Brent's algorithm using our quantile bounds as the box constraints. Furthermore, we adaptively set the termination conditions xtol and ftol to be on the same order of magnitude as the error in quadrature sparsification from §4.2. This greatly accelerates convergence in practice. A performance comparison between quantile bounds outlined in this section can be found in §5.

### 4.4 Fast Leave-One-Out-Cross-Validation (LOOCV)

LOOCV is a standard measure of model fit: in practice, it is most commonly used for tuning hyperparameters and for model selection. While fast LOOCV schemes are known for GP regression, it is less straightforward to perform LOOCV on BTG. In particular, the computational difficulty lies in two LOOCV sub-problems: a generalized least squares problem and principle sub-matrix determinant computation. These correspond to the terms in the BTG likelihood function and the BTG conditional posterior in Equation 6. Being Bayesian about covariance and transform hyperparameters introduces additional layers of cost: LOOCV must be repeated at each quadrature node in the hyperparameter space. This further motivates the need for an efficient algorithm.

For notational clarity, let $(-i)$ denote the omission of the $i$th point. For a kernel matrix, this means deletion of the $i$th row and column; for a vector, this indicates the omission of the $i$th entry. We seek to compute the mean $m_{\theta,\lambda}^{(-i)}$ and standard deviation $\sigma_{\theta,\lambda}^{(-i)} = \left( C_{\theta,\lambda}^{(-i)} q_{\theta,\lambda}^{(-i)} \right)^{-1/2}$ of the t-distributions (Equations 5) for each submodel, obtained by leaving out the $i$th training point. Specifically, computing $\{q_{\theta,\lambda}^{(-i)}\}_{i=1}^{n}$ entails solving the generalized least squares problems for $i = 1, \ldots, n$:

$$\arg\min_{\beta^{(-i)}} \left\| Y^{(-i)} - M_X^{(-i)} \beta^{(-i)} \right\|_{K_X^{(-i)}}^2,$$

Table 2: Elementary Transformations: analytic function forms and parameter constraints. Parameters are assumed to be in $\mathbb{R}$ unless stated otherwise.

| Name ‖ | $g(y)$ | Req. | Num. Params |
|---|---|---|---|
| Affine | $a + by$ | $b > 0$ | 1 |
| ArcSinh | $a + b \,\mathrm{asinh}\left(\dfrac{y - c}{d}\right)$ | $b, d > 0$ | 4 |
| SinhArcSinh | $\sinh(b \,\mathrm{asinh}(y - a))$ | $b > 0$ | 2 |
| Box-Cox | $\begin{cases} \dfrac{y^{\lambda} - 1}{\lambda} & \text{if } \lambda > 0 \\ \log(y) & \text{if } \lambda = 0 \end{cases}$ | $\lambda \geq 0$ | 1 |

where $Y = g \circ f_X$. In addition, computing $C_{\theta,\lambda}^{(-i)}$ and $m_{\theta,\lambda}^{(-i)}$ entails solves with $K_X^{(-i)}$, which naively takes $\mathcal{O}(n^3)$ per sub-problem. Therefore, the BTG LOOCV proceedure naively takes $\mathcal{O}(n^4)$ total time.

We develop an $\mathcal{O}(n^3)$ fast LOOCV algorithm for BTG using three building blocks: fast determinant computations (Proposition 4.6), fast abridged linear system solves (Proposition 4.7) and fast rank-one $\mathcal{O}(p^2)$ Cholesky down-dates (Proposition 4.8). We refer to Stewart (1998) for the rank-1 Cholesky downdate algorithm. For algorithm details as well as proofs, see Appendix B. The scaling behavior of our fast LOOCV algorithm is shown in Figure 4.

**Proposition 4.6** (Determinant of a Principal Minor).

$$\det\left(\Sigma^{(-i)}\right) = \det(\Sigma)\left(e_i^T \Sigma^{-1} e_i\right).$$

**Proposition 4.7** (Abridged Linear System). *Let $K \in \mathbb{R}^{n \times n}$ be of full rank, and let $c, y \in \mathbb{R}^n$ satisfy $Kc = y$. Then if $r_i = c_i / e_i^T K^{-1} e_i$, we have:*

$$c^{(-i)} = (K^{(-i)})^{-1} y^{(-i)} = c - r_i K^{-1} e_i.$$

**Proposition 4.8** (Rank one matrix downdate). *If $X \in \mathbb{R}^{n \times m}$ with $m < n$ has full column rank and $\Sigma$ is a positive definite matrix in $\mathbb{R}^{n \times n}$, then we have*

$$X^{(-i)^T} \Sigma^{(-i)^{-1}} X^{(-i)} = X^T \left(\Sigma^{-1} - \frac{\Sigma^{-1} e_i e_i^T \Sigma^{-1}}{e_i^T \Sigma^{-1} e_i}\right) X,$$

*where $\Sigma^{(-i)} \in \mathbb{R}^{(n-1) \times (n-1)}$ is the $(i, i)$th minor of $\Sigma$ and $e_i$ is the $i$th canonical basis vector.*

### 4.5 Transformations

The original BTG model of Oliveira et al. (1997) uses the Box-Cox family of power transformations and places an uniform prior on $\lambda$. Recent research has greatly expanded the set of flexible transformations available. Snelson et al. (2004) uses a sum of $\tanh(\cdot)$ transforms in the WGP model and Rios & Tobar (2019) composes various transformations to provide a flexible compositional framework in the CWGP model.

We apply BTG with more elementary transformations and compositions thereof, summarized in Table 2. As we show in §5, these compositions have greater expressive power and generally outperform single transformations, at the expense of greater computational overhead.

## 5 Experiments

This section contains:

- Experimental details necessary to reproduce our experiments, including the datasets we used as well as the decisions we made regarding the BTG and GP models (such as the kernel).

Figure 2: Plots of marginal log likelihoods for a SinhArcSinh-transformed WGP and corresponding training data. In the first four columns, $a, b$ are transformation hyperparameters, and $\theta, \sigma^2$ are kernel hyperparameters; the optimal values of marginal log likelihoods are marked with a red star. The last column shows the underlying function (dashed curves) and the training data (red dots). The top row represents a data-sparse setting with 5 training points, while the bottom row represents a data-rich setting with 30 training points. We can see that in the data-sparse setting, optimal marginal log likelihood values with respect to $\theta$ and $\sigma^2$ are clearly not as well defined as the data-rich setting.

- A small example highlighting that when data is sparse, the likelihoods for transformations and kernel hyperparameters can be very flat, thus demonstrating the need for BTG in these scenarios.

- Experiments to validate the efficiency of our computational techniques.

- Thorough regression experiments, which demonstrate BTG's strong empirical performance when compared to appropriately selected baselines. We also provide timing results of our fast BTG implementation, showing that BTG has a comparable computational cost to WGP.

We hope these experiments provide a thorough and nuanced picture that highlights *when and why* BTG can perform better than it's GP counterparts. All methods possess their strengths and weaknesses, and BTG is not an exception.

## 5.1 Motivation for the Bayesian Approach

## 5.2 Experiment Details

We provide the datasets we use, the performance metrics we measure and additional implementation details necessary to reproduce our experiments in this subsection.

**Two synthetic datasets: `IntSine` and `SixHumpCamel`.** The `IntSine` dataset, also used by Lázaro-Gredilla (2012), is sampled from a rounded 1-dimensional sine function with Gaussian noise of a given variance. The training set is comprised of 51 uniformly spaced samples on $[-\pi, \pi]$. The testing set consists of 400 uniformly spaced points on $[-\pi, \pi]$. The `SixHumpCamel` function is a 2-dimensional benchmark optimization function usually evaluated on $[-3, 3] \times [-2, 2]$ (Molga & Smutnicki, 2005). We shift its values to be strictly positive. The training set is comprised of 50 quasi-uniform samples, i.e., a 2d Sobol sequence, from $[-1, 1] \times [-2, 2]$. The testing set consists of 400 uniformly distributed points on the same domain.

**Three real datasets: `Abalone`, `WineQuality` and `Creep`.** `Abalone` is an 8-dimensional dataset, for which the prediction task is to determine the age of an abalone using eight physical measurements (Dua & Graff, 2017). The `WineQuality` dataset has 12-dimensional explanatory variables and relates the quality of wine to input attributes (Cortez et al., 2009). The `Creep` dataset is 30-dimensional and relates the creep rupture

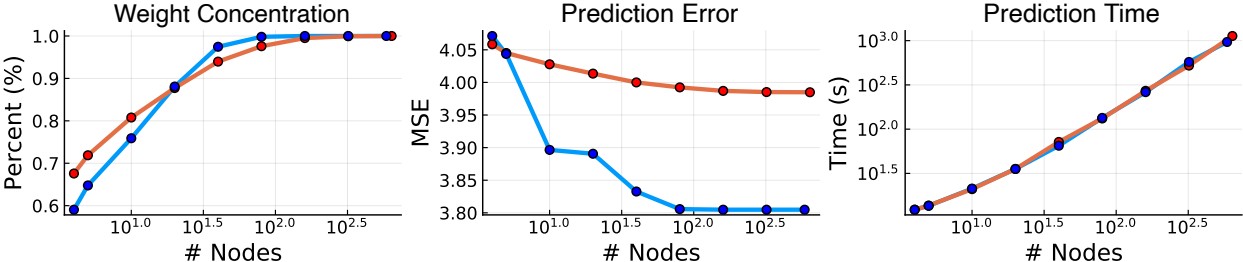

Figure 3: Comparison of sparse grid quadrature (blue) and QMC (red). Weights are ordered by decreasing magnitude. (*Left*) Number of nodes versus percent of total mass they comprise (*Middle*) Number of nodes versus BTG prediction MSE (*Right*) Number of nodes versus prediction time. We can see that sparse grid quadrature has a prediction error that decreases much faster than QMC.

stress (in MPa) for steel to chemical composition and other features (Cole et al., 2000). To simulate data-sparse training scenarios, we randomly select training samples of size 30, 200, and 100 from `Abalone`, `WineQuality`, and `Creep`, respectively, and test on 500, 1000 and 1500 out-of-sample points.

### 5.2.1 Sparse Grids vs QMC

**Performance Metrics:** Our empirical experiments involve three loss functions to evaluate model performance: root mean squared error (RMSE), mean absolute error (MAE) and mean negative log predictive density (NLPD). Let $\{f^*(\boldsymbol{x_i})\}_{i=1}^n$ be true test labels and $\{\hat{f}(\boldsymbol{x_i})\}_{i=1}^n$ be predictions, which are taken to be predictive medians in WGP and BTG. The predictive median is also used by Snelson et al. (2004). Let $\bar{p}$ be the predictive distribution from model. The RMSE, MAE and NLPD are defined as follows

$$\text{RMSE} = \left(\frac{1}{n}\sum_{i=1}^n (f^*(\boldsymbol{x_i}) - \hat{f}(\boldsymbol{x_i}))^2\right)^{\frac{1}{2}},$$

$$\text{MAE} = \frac{1}{n}\sum_{i=1}^n \left|f^*(\boldsymbol{x_i}) - \hat{f}(\boldsymbol{x_i})\right|,$$

$$\text{NLPD} = -\frac{1}{n}\sum_{i=1}^n \log(\bar{p}(f^*(\boldsymbol{x_i}))).$$

**Implementation:** We run all experiments using our Julia software package, which supports a variety of models (WGP, CWGP and BTG) and allows for flexible treatment of hyperparameters. We also implement several single and composed transformations. For MLE optimization, we use the L-BFGS algorithm from the Julia `Optim` package (Mogensen & Riseth, 2018).

**Kernel:** We used the squared exponential (RBF) kernel for all experiments:

$$k_\theta(\boldsymbol{x}, \boldsymbol{x'}) = \frac{1}{\tau^2}\exp\left(-\frac{1}{2}\|\boldsymbol{x} - \boldsymbol{x'}\|_{D_\theta^{-2}}^2\right) + \sigma^2\delta_{\boldsymbol{xx'}}.$$

**Model:** To model observation input noise for BTG, we add a regularization term to make the analytical marginalization of mean and precision tractable. We also assume the constant covariate $m(\boldsymbol{x}) = 1_n$ in the BTG model, and normalize observations to the unit interval. We assume the constant mean field for both BTG and WGP.

Our motivation for using a Bayesian approach like BTG rather than a MLE-based one like WGP is that the maximum value of the marginal log likelihood in a data-sparse setting is not as well-defined as that in a data-rich setting. We demonstrate this motivation using a 1D toy example. We examine the marginal log

Table 3: Compute time (seconds) for computing predictive median and credible intervals using no quantile bound, convex hull quantile bound, and singular weight quantile bound. Results are averaged over 10 trials. We find that both bounds significantly reduce the total time used to compute the predictive median and credible intervals —though the convex hull bound performs better on average than the singular weight bound, likely because the former is often a better bound than the latter.

|  | Total (s) | Median (s) | Credible Interval (s) |
|---|---|---|---|
| Baseline | 13.0 | 3.24 | 7.87 |
| Convex Hull | 6.21 | 1.11 | 3.19 |
| Singular Weight | 11.0 | 2.52 | 6.54 |

likelihoods of transformation and kernel hyperparameters in the WGP model in Figure 2. We consider a 1D synthetic function, the 1D Levy function:

$$f(\boldsymbol{x}) = \sin^2\left(\frac{(\boldsymbol{x}+3)\pi}{4}\right) + \left(\frac{\boldsymbol{x}-1}{4}\right)^2\left(1 + \sin^2\left(\frac{(\boldsymbol{x}+3)\pi}{2}\right)\right).$$

As a toy example, we use $n = 5$ training data for a data-sparse setting and use $n = 30$ for a data-rich setting. We use a SinhArcSinh transformation for both settings, which is parameterized by two parameters $a$ and $b$ (see Table 2). We thus have four total hyperparameters $(a, b, \theta, \sigma)$, and we plot their marginal log likelihoods in Figure 2 for both the data-sparse and data-rich setting. We also identify the optimal hyperparameters $(a^*, b^*, \theta^*, \sigma^*)$ using MLE, which we plot with a red star.

We observe that in the data-sparse setting, the marginal log likelihoods tend to be quite flat, even on a log scale, which is especially true of $\theta$ and $\sigma^2$. This means that many possible hyperparameters explain the data, and consequently that an MLE-based model (which selects only one hyperparameter value) will likely be a poor fit. This also suggests that a fully Bayesian approach—which is what BTG does—could be more appropriate in the data-sparse setting than MLE estimation.

In the data-rich setting, the distribution of $\theta$ and $\sigma^2$ are tightly concentrated, and thus the MLE approach does make sense.

### 5.3 Scaling Experiments

We demonstrate the efficiency of the proposed computational techniques discussed in §4. First, we illustrate our quadrature sparsification scheme discussed in §4.2 on two different quadrature rules, and compare the performance under varying sparsification levels. Second, we evaluate the fast quantile bound for root-finding used in BTG prediction proposed in §4.3 by comparing the total time cost of BTG with WGP. Last but not least, we verify the complexity improvement of the proposed fast cross validation.

We compare sparse grid and QMC quadrature rules under our quadrature sparsification framework. We use the `SixHumpCamel` dataset with 30 training data points and 100 testing data points as a toy problem. We train BTG with the composed transformation Affine-SinhArcSinh. The hyperparameter space is 7-dimensional. In our experiment, we begin with a handful of quadrature nodes, and gradually extend this set to the entire quadrature grid. As more and more quadrature nodes are incorporated into the approximate integral (the approximate posterior distribution), we record the percentage of total weight that is used, and then compare how the prediction MSE and the prediction time cost vary. Intuitively, increasing quadrature nodes would reduce the prediction MSE and increase the prediction time cost. To better study our quadrature sparsification framework and balance the prediction accuracy and cost, we closely compare such phenomenon on the two quadrature rules allowed in BTG. Results are plotted in Figure 3.

From Figure 3 we observe that the sparse grid quadrature rule yields lower prediction MSE: QMC converges to an MSE of 3.99, while sparse grid converges to an MSE of 3.80. We also observe that the sub-grids have similar *weight concentration*, which we showed was a proxy for quadrature approximation error in §4. Therefore, as the mass of the dropped weights falls below 0.1, the error in the integration scheme can

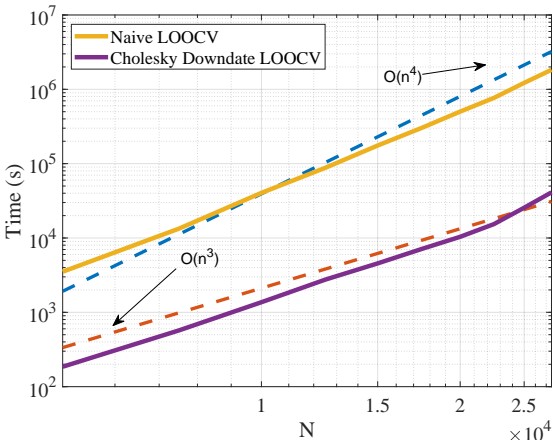

Figure 4: Compare the LOOCV timing of computing posterior distribution parameters of $n$ sub-models with the proposed Cholesky downdate and without. It is verified that the naive way has a $\mathcal{O}(n^4)$ cost while our proposed Cholesky downdate reduce the cost, by a factor of $n$, to $\mathcal{O}(n^3)$.

increasingly be attributed to the error in the quadrature rule itself as opposed to sparsification. Since the error of the sparse grid rule decays faster than that of QMC, we expect sparse grid prediction error to also decay more quickly and this trend is supported by Figure 3. We empirically find that the joint-likelihood function is sufficiently smooth, so that sparse grids are effective. Finally, we confirm that inference time scales linearly with the number of quadrature nodes.

### 5.3.1 Quantile Bounds Speedup

To assess the effectiveness of quantile bounds for root-finding, we record BTG prediction times using the convex hull bound and singular weight bound proposed in §4.3. We set up a problem using the `Levy1D` dataset Molga & Smutnicki (2005) and 200 training points. The tolerance for the root-finding Brent's algorithm is set to be $10^{-3}$. We record the total time cost of prediction, the time cost of computing medians and time cost of computing credible intervals. Intuitively, using a more precise bound in a root-finding algorithm would greatly improve convergence speed, and this is verified with results in Table 3.

We find that the convex hull bound decreases the overall computational overhead by a factor of at least two. The convex hull bound outperforms the singular weight bound for finding credible intervals and in overall time, but the singular weight bound was faster for finding the median in many scenarios.

### 5.3.2 Fast Cross Validation

To assess our fast LOOCV proposed in §4.4, we infer the sub-model posterior distribution moments on a toy problem in two different ways: with and without using Cholesky rank-one downdates on $R_X$ to compute the Cholesky factors for $R_X^{(-i)}$. We plot timing results in Figure 4, which confirm that our $\mathcal{O}(n^3)$ method scales significantly better than the naive $\mathcal{O}(n^4)$ method. We note that these timings are generated using exact GP inference, and that more generally our fast cross validation is $\mathcal{O}(n)$ faster than standard cross validation. We note that this speed-up applies not only to the typical $\mathcal{O}(n^3)$ cost of dense GP inference, but also any other scalable GP method, such as variational GPs which would be a $\mathcal{O}(n^2)$ cost instead.

### 5.4 Regression and Uncertainty Quantification Experiments

Our efficient algorithms allow us to test BTG on a set of synthetic and real-world problems. We consider 5 datasets: `IntSine`, `SixHumpCamel`, `Abalone`, `Wine`, and `Creep` with dimension ranging from 1 to 30. The total dimension of the hyperparmeter space is further increased by transform parameters by as much as 8.

Table 4: A comprehensive set of RMSE and NLPD results for prediction using the `IntSine`, `SixHumpCamel`, `Abalone`, `Wine` and `Creep` datasets. We generally kept the training sets small, to highlight BTG's advantages in the data-sparse regime, though for the `IntSine` function there was a good amount of data. We compare the following models[3]: GP, DeepGP, WGP, CWGP, BTG, and use the following transformations transformations: I: Identity, A: ArcSinh, SA: SinhArcSinh, BC: BoxCox, L: affine. The best method is bolded; we find that BTG has lower RMSE and NLPD than the other models, though unsurprisingly, different transformations are more suitable for different datasets.

| | IntSine | | Camel | | Abalone | | Wine | | Creep | |
|---|---|---|---|---|---|---|---|---|---|---|
| | RMSE | NLPD | RMSE | NLPD | RMSE | NLPD | RMSE | NLPD | RMSE | NLPD |
| GP | 0.227 | -1.179 | 2.003 | 11.20 | 3.290 | 1.120 | 1.994 | -0.074 | 37.88 | -1.189 |
| DeepGP | 0.241 | 0.013 | **1.344** | 1.761 | 3.212 | 3.988 | 0.824 | 1.377 | 91.95 | 20.39 |
| WGP-A | 0.179 | -1.392 | 2.055 | 44.10 | 3.097 | 1.105 | 0.811 | -1.108 | 35.48 | -1.421 |
| WGP-SA | 0.172 | -1.693 | 2.012 | 45.10 | 2.992 | 4.879 | 0.809 | -1.269 | 61.62 | 1.001 |
| WGP-BC | 0.184 | -1.371 | 1.964 | 44.70 | 2.826 | 0.837 | 1.045 | -0.186 | 40.38 | -1.357 |
| CWGP-L-SA | 0.172 | -1.693 | 2.055 | 13.80 | 3.117 | 3.444 | 0.808 | -1.415 | 39.71 | -1.134 |
| CWGP-A-BC | 0.174 | -1.457 | 1.960 | 37.90 | 3.088 | 1.151 | 0.808 | **-1.416** | 37.89 | -1.399 |
| BTG-I | 0.169 | -1.429 | 1.820 | 2.342 | 2.804 | -0.070 | 0.808 | -1.356 | 38.11 | -1.273 |
| BTG-A | 0.168 | -1.443 | 1.827 | 2.331 | 2.890 | -0.174 | 0.807 | -1.360 | 38.65 | -1.239 |
| BTG-SA | 0.165 | -1.759 | 1.801 | 0.907 | **2.791** | -0.123 | 0.820 | -0.856 | 91.69 | -0.358 |
| BTG-BC | 0.170 | -1.439 | 1.675 | **0.659** | 3.225 | 0.110 | 0.808 | -1.358 | 39.10 | -1.332 |
| BTG-L-SA | **0.145** | **-1.781** | 1.673 | 0.730 | 2.871 | -0.023 | 0.809 | -1.359 | 91.83 | -0.374 |
| BTG-A-BC | 0.159 | -1.516 | 1.828 | 2.330 | 2.832 | **-0.299** | **0.802** | -1.361 | **35.25** | **-1.433** |

We compare with a standard GP model, WGP models and CWGP models using transformations[4] and their compositions from Table 2.

For each of these datasets, we perform both regression and uncertainty quantification (UQ) by computing the RMSE and NLPD, which we record in Table 4. Our two metrics evaluate both the predictive mean and predictive variance, respectively: RMSE is the root mean squared error metric evaluating predictive mean, and NLPD is the mean negative log predictive density metric evaluating both the predictive mean and predictive variance (uncertainty). For space's sake, we record additional performance metrics such as MAE in the appendix (Table 6).

We also compare end-to-end inference time cost in Table 5 to give the reader a better idea of the computational overhead associated with BTG. This compute time generally depends on the dimension of the integral, i.e., the total number of hyperparameters that we must marginalize out in the fully Bayesian approach, and thus the numbers vary by a bit depending on the dataset.

Our results are largely favorable. We find that composed BTG models tend to outperform single-transformation BTG models, which themselves tend to outperform their GP, WGP, and CWGP counterparts. The DeepGP model performs poorly in most of these problems due to the lack of enough data, which would otherwise allow it to overcome the approximation error associated with variational inference. More generally, BTG outperforms other baselines on almost all problems for both regression and uncertainty quantification (UQ), and quite convincingly in certain cases —for example, BTG achieves an NLPD around two orders of magnitude better than existing models in the Camel dataset.

[5] These results demonstrate the improved flexibility afforded by layered transformations, and is evidence of the superior performance possible with a fully Bayesian approach on small to medium sized datasets.

---

[4]We omit the tanh($\cdot$) transform used in the original WGP paper (Snelson et al., 2004) since Rios & Tobar (2019) shows that the elementary transforms in Table 2 are competitive with tanh($\cdot$).

[5]We tried to compare against BWGP, but could not find a code release available. We suspect BWGP to possess similar performance to the DeepGP, as both use variational inference and highly flexible families of warping functions.

Table 5: End-to-end regression times for `SixHumpCamel`, `Abalone` and `Wine` datasets, using the following models: GP, WGP, CWGP, BTG, and the following transformations: SA:SinhArcSinh, BC:BoxCox, L:affine. We note that the timings for BTG depends on the number of quadrature nodes it uses, but we made sure that these settings were consistent with those of Table 4, in which BTG outperforms GP, WGP, and CWGP.

|  | SixHumpCamel | | Abalone | | Wine | |
|---|---|---|---|---|---|---|
|  | Dim | Time (min) | Dim | Time (min) | Dim | Time (min) |
| WGP-BC | 5 | 0.68 | 10 | 1.28 | 14 | 1.52 |
| WGP-SA | 6 | 0.78 | 11 | 1.22 | 15 | 1.60 |
| CWGP-L-SA | 8 | 1.08 | 13 | 1.48 | 17 | 2.50 |
| CWGP-A-BC | 9 | 1.14 | 15 | 1.56 | 18 | 2.82 |
| BTG-BC | 4 | 1.10 | 9 | 1.02 | 13 | 1.40 |
| BTG-SA | 5 | 0.95 | 10 | 1.04 | 14 | 1.29 |
| BTG-L-SA | 7 | 1.74 | 12 | 1.11 | 16 | 1.79 |
| BTG-A-BC | 8 | 1.65 | 14 | 1.09 | 17 | 1.87 |

Additionally, due to our efficient numerical methods, the end-to-end inference time of BTG is generally comparable to other baselines –the compute time of BTG is only a bit higher than that of WGP.

## 6 Conclusion

In this paper, we revisited the Bayesian transformed Gaussian (BTG) model, which is a variant of Bayesian trans-Kriging. BTG can be seen as a fully Bayesian treatment of input-warped GPs, and we believe it fills an important niche in the GP literature, which has yet not explored the combination of transformed GPs and Bayesian inference.

We have presented a set of efficient methods for efficiently computing with the BTG model. These methods combine sparse grid quadrature, quadrature sparsification, and tight quantile bounds significantly reduces the expense of computing BTG's predictive median—in certain cases rivaling even the speed of MLE—without degrading prediction accuracy. Furthermore, we proposed a fast LOOCV algorithm for BTG for model selection and assessing model fit. An advantage of our framework is that it allows the practitioner to control the trade-off between the speed and accuracy of the Bayesian approach by modulating the sparsification of the grid and tolerance of the quantile-finding routine.

Using these efficient methods, we were able to compute a set of experiments suggesting that BTG demonstrates superior performance over its comparable models in low-data and medium-data regimes, where hyperparameters are likely under-specified.

In future work, we would like to combine our approach with approximate GP inference to further improve computational efficiency. Furthermore, though our sparse quadrature rules allow us to go to higher dimensions, we will still eventually hit a limit (depending on the problem, perhaps around 50); to go to dimensions in the hundreds or thousands, additional assumptions and methodologies will be needed. Lastly, we would like to apply BTG to Bayesian optimization, which is one application we believe to be particularly suitable for it, since we expect data to be sparse.

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

## A   Methodology

### A.1   Quadrature Sparsification

We assume the posterior cdf is the mixture of cdfs $F(x) = \sum_{i=1}^{M} w_i f_i(x)$, $0 \leq f_i(x) \leq 1$, and $f_i(x)$ are monotone increasing for $i = 1, \ldots, M$. Assume the weights $\{w_i\}_{i=1}^{M}$ are decreasingly ordered by magnitude from 1 to $M$. We consider the quantiles of the approximant $F_k(x)$, a truncated and re-scaled $F(x)$.

**Lemma A.1.** *Define $k$ to be the smallest integer such that $\sum_{i=1}^{k} w_i \geq 1 - \epsilon$. Then define the scaled, truncated mixture*

$$F_k(x) := \frac{1}{c} \sum_{i=1}^{k} w_i f_i(x), \quad c := \sum_{i=1}^{k} w_i$$

*We have*

$$|F(x) - F_k(x)| \leq 2\epsilon.$$

*Proof.* Let $R_k(x) = |F_k(x) - F(x)|$. We have

$$R_k(x) = \left| \left( \frac{1}{c} - 1 \right) \sum_{i=1}^{k} w_i f_i(x) + \sum_{i=k+1}^{M} w_i f_i(x) \right|$$

$$\leq \left| \left( \frac{1}{c} - 1 \right) \sum_{i=1}^{k} w_i f_i(x) \right| + \left| \sum_{i=k+1}^{M} w_i f_i(x) \right|$$

$$\leq \frac{1 - c}{c} \sum_{i=1}^{k} w_i f_i(x) + 1 - c$$

$$\leq 2(1 - c) \leq 2\epsilon.$$

□

**Proposition A.1** (Error Bound for Positive Weights). *For any $\epsilon \in (0, 1)$, let $k$ be the smallest integer such that $\sum_{i=1}^{k} w_i \geq 1 - \epsilon$. Then define the scaled, truncated mixture*

$$F_k(x) := \frac{1}{c} \sum_{i=1}^{k} w_i f_i(x), \quad c := \sum_{i=1}^{k} w_i.$$

*Let $p \in (0, 1)$ and assume that $p \pm 2\epsilon \in (0, 1)$. Then we have the bound:*

$$F^{-1}(p - 2\epsilon) \leq F_k^{-1}(p) \leq F^{-1}(p + 2\epsilon).$$

*Proof.* Let $F_k(x^*) = p$. Then $|p - F_k(x^*)| \leq 2\epsilon$, so

$$p - 2\epsilon \leq F(x^*) \leq p + 2\epsilon$$

It follows that

$$F^{-1}(p - 2\epsilon) \leq x^* \leq F^{-1}(p + 2\epsilon).$$

□

**Proposition A.2** (Error Bound for Negative Weights). *Let $F(x)$ be defined as before, except each $w_i$ is no longer required to be positive. Consider the split $F(x) = F_{M'}(x) + R_{M'}(x)$, where $F_{M'}(x) = \sum_{i=1}^{M'} w_i f_i(x)$ and $R_{M'}(x) = \sum_{i=M'+1}^{M} w_i f_i(x)$. Then for any $x$, we have $R_{M'}(x) \in [\epsilon_-, \epsilon_+]$, where the epsilons are defined as the sum of positive (resp. negative) weights of $R_{M'}(x)$*

$$\epsilon_- = \sum_{i=M'+1}^{M} [w_i]_- \leq 0 \, , \, \epsilon_+ = \sum_{i=M'+1}^{M} [w_i]_+ \geq 0.$$

*Then we bound $F^{-1}(p)$ as follows:*

$$F^{-1}(p + \epsilon_-) \leq F_{M'}^{-1}(p) \leq F^{-1}(p + \epsilon_+).$$

*Proof.* Let $F_{M'}(x^*) = p$. Then we can wrote $F(x^*) = p + R_{M'}(x^*)$. Since $\epsilon_- \leq R_{M'}(x^*) \leq \epsilon_+$, it follows that

$$p + \epsilon_- \leq F(x^*) \leq p + \epsilon_+$$

from which the result follows. $\qquad\square$

**Proposition A.3** (Error Bound at a quantile). *Let $F(x)$ be defined as before, $\epsilon_1, \epsilon_2 \in (0, 1)$, and $F_k(x)$ be an approximate to $F(x)$ such that $F^{-1}(p - \epsilon_1) \leq F_k^{-1}(p) \leq F^{-1}(p - \epsilon_2)$ for some $p \in (0, 1)$. Assuming $p - \epsilon_1, p + \epsilon_2 \in (0, 1)$, we have the following error bound at a quantile,*

$$\left| F_k^{-1}(p) - F^{-1}(p) \right| \leq \epsilon \max_{\xi \in (p - \epsilon_1, p + \epsilon_2)} \left| \frac{dF^{-1}}{dx}(\xi) \right|,$$

*where $\epsilon = \max\{\epsilon_1, \epsilon_2\}$.*

*Proof.* We have

$$
\begin{aligned}
&\left| F_k^{-1}(p) - F^{-1}(p) \right| \\
&\leq \max\{|F^{-1}(p - \epsilon_1) - F^{-1}(p)|, |F^{-1}(p - \epsilon_2) - F^{-1}(p)|\} \\
&\leq \max\{\epsilon_1, \epsilon_2\} \max_{\xi \in (p - \epsilon_1, p + \epsilon_2)} \left| \frac{dF^{-1}}{dx}(\xi) \right|.
\end{aligned}
$$

$\qquad\square$

## A.2 Quantile Bounds

**Proposition A.4** (Convex Hull). *Let $F(x)$ be defined as before with $w_i > 0$ and $\sum_{i=1}^{M} w_i = 1$. Then*

$$\min_i f_i^{-1}(p) \leq F^{-1}(p) \leq \max_i f_i^{-1}(p).$$

*Proof.* Assume for the sake of contradiction that $F^{-1}(p) > \max_i f_i^{-1}(p)$. Let $k = \arg\max_i f_i^{-1}(p)$. Using the fact that $f_k^{-1}(p) \geq f_j^{-1}(p)$ for any $j \neq k$, we have $f_j(f_j^{-1}(p)) \geq p$ for any $j \neq k$. Therefore, we have

$$
\begin{aligned}
p &> F(f_k^{-1}(p)) \\
&= \sum_{j=1}^{M} w_j f_j(f_k^{-1}(p)) \\
&= w_k p + \sum_{j \neq i} w_j f_j(f_k^{-1}(p)) \\
&\geq w_k p + (1 - w_k)p = p,
\end{aligned}
$$

which leads to a contradiction. The lower bound is analogous. $\qquad\square$

**Proposition A.5** (Singular Weight). *Let $F(x)$ be defined as before with $w_i > 0$ and $\sum_{i=1}^{M} w_i = 1$. Let $\overline{w}_i = 1 - w_i$. Then*

$$\max_{p - \overline{w}_i \geq 0} f_i^{-1}(p - \overline{w}_i) \leq F^{-1}(p) \leq \min_{p + \overline{w}_i \leq 1} f_i^{-1}(p + \overline{w}_i).$$

*Proof.* Assume for sake of contradiction that

$$F^{-1}(p) > f_i^{-1}(p + \overline{w}_i)$$

Then

$$p > F\left(f_i^{-1}(p + \overline{w}_i)\right)$$

$$= \sum_{j=1}^{N} w_j f_j \left(f_i^{-1}(p + \overline{w}_i)\right)$$

$$= w_i(p + \overline{w}_i) + \sum_{j \neq i} w_j f_j \left(f_i^{-1}(p + \overline{w}_i)\right)$$

$$\geq w_i(p + \overline{w}_i)$$

However this implies that $0 > \overline{w}_i(w_i - p)$, which is a contradiction because $1 \geq w_i$ and $w_i \geq p$ (by the assumption that $p + \overline{w}_i \leq 1$). The lower bound is analogous. $\qquad\square$

## B    Fast Cross Validation

In this section, we discuss results leading up to $\mathcal{O}(n^3)$ LOOCV algorithms for BTG, which are given by Algorithm 1 and Algorithm 2. Naively, the BTG LOOCV procedure has $\mathcal{O}(n^4)$ time cost, due to the costs associated with solving generalized least squares problems related by single-point deletion and evaluating determinants of principle submatrices. We present relevant propositions used to solve these LOOCV subproblems efficiently in §B.1, and derive our full algorithm in §B.2.

**Notation** Let $f_X, M_X, K_X, \boldsymbol{x}, \sigma_{\theta,\lambda}, q_{\theta,\lambda}, m_{\theta,\lambda}$ and $C_{\theta,\lambda}$ be defined same as in the paper. As before, we use the $(-i)$ notation to represent to omission of information from the $i$th data point. For the BTG LOOCV problem, we consider the $n$ submodels $\{\text{Model}^{(-i)}\}_{i=1}^n$ trained on $\{\mathbf{x}^{(-i)}, f_X^{(-i)}, M_X^{(-i)}\}_{i=1}^n$: the location-covariate-label triples obtained by omitting data points one at a time. We wish to efficiently compute the posterior predictive distributions of all $n$ submodels indexed by $i \in \{1, ..., n\}$,

$$p\left(f(\mathbf{x}^{(-i)}) \mid f_X^{(-i)}\right) \propto \sum_{j=1}^{M} w_j L_j J_j p(\theta_j) p(\lambda_j), \tag{9}$$

where $L_j = p\left(g_{\lambda_j}(f(\boldsymbol{x}^{(-i)})) \mid \theta_j, \lambda_j, f_X^{(-i)}\right)$ and
$J_j = p\left(f_X^{(-i)} | \theta_j, \lambda_j\right)$ for $j \in \{1, 2, \ldots, M\}$.

Recall that in Equation 9, $p\left(g_\lambda(f(\boldsymbol{x})) \mid \theta, \lambda, f_X\right)$ is the probability density function of the $t$-distribution $T_{n-p}(m_{\theta,\lambda}, (q_{\theta,\lambda} C_{\theta,\lambda})^{-1})$ and $p(f_X|\theta, \lambda)$ is the likelihood of data given hyperparameters.

**Problem Formulation** We have to efficiently compute the parameters that of the posterior mixture of t-distributions in Equation 9:

$$\{\text{TParameters}^{(-i)}\}_{i=1}^n := \left\{ m_{\theta_i,\lambda_i}^{(-i)}, q_{\theta_i,\lambda_i}^{(-i)}, C_{\theta_i,\lambda_i}^{(-i)} \right\}_{i=1}^n$$

For definitions of these quantities, we refer to the main text. We instead emphasize here that solving for $q_{\theta,\lambda}^{(-i)}$ entails solving perturbed generalized least squares problems and that solving for $m_{\theta,\lambda}^{(-i)}$ and $C_{\theta,\lambda}^{(-i)}$ entail solving perturbed linear systems.

For the likelihood term in Equation 9, we have

$$p(f_X|\theta, \lambda) \propto \left|\Sigma_\theta\right|^{-1/2} \left|M_X^T \Sigma_\theta^{-1} M_X\right|^{-1/2} q_{\theta,\lambda}^{(-(n-p)/2)},$$

hence we are interesting in computing the following for $i \in \{1, 2, \ldots, n\}$:

$$\text{Det}^{(-i)} = \left\{ \left|\Sigma_\theta^{(-i)}\right|, \left|(M_X^{(-i)})^T \Sigma_\theta^{(-i)} M_X^{(-i)}\right| \right\}. \tag{10}$$

The perturbed least squares problems and linear systems can be solved independently in $\mathcal{O}(n^3)$ time, hence a naive LOOCV procedure would take $\mathcal{O}(n^4)$ time. However, using matrix decompositions, we can improve this to $\mathcal{O}(n^3)$ total time.

**Algorithms** Algorithms 1 and 2 are used for efficiently computing $\{\text{TParameters}^{(-i)}\}_{i=1}^n$ and $\{\text{Det}^{(-i)}\}_{i=1}^n$ for fixed hyperparameters $(\theta, \lambda)$. The total time complexity is $\mathcal{O}(n^3)$, because the dominant costs are precomputing a Cholesky factorization for a kernel matrix and repeating $\mathcal{O}(n^2)$ operations across $n$ submodels.

---

**Algorithm 1** T-Distributions of Sub-Models

---

**Inputs** $Y = g_\lambda \circ f_X$, $M_X$, $K_X$, $\boldsymbol{x}$
**Outputs:** $\{m_{\theta,\lambda}^{(-i)}\}_{i=1}^n$, $\{q_{\theta,\lambda}^{(-i)}\}_{i=1}^n$, $\{C_{\theta,\lambda}^{(-i)}\}_{i=1}^n$
Pre-compute $R, R_X$, and $\hat{\boldsymbol{x}}$, where $R^T R = K_X$, $R_X^T R_X = M_X^T K_X^{-1} M_X$, $\hat{\boldsymbol{x}} = K_X^{-1} Y$
**for** $i = 1 \dots n$ **do**
   $\ell_i = K_X^{-1} e_i / |e_i^T K^{-1} e_i|$
   $R_X^{(-i)} \leftarrow \text{Downdate}(R_X, \ell_i)$     (Proposition B.3)
   $r_i \leftarrow Y_i / \|R^{-T} e_i\|_2^2$
   $\hat{\boldsymbol{x}}^{(-i)} \leftarrow \hat{\boldsymbol{x}} - r_i R^{-1}(R^{-T} e_i)$     (Proposition B.2)
   $\beta_{\theta,\lambda}^{(-i)} \leftarrow \left(R_X^{(-i)}\right)^{-1}\left(R_X^{(-i)}\right)^{-T} M_X^{(-i)} \hat{\boldsymbol{x}}^{(-i)}$
   $r^{(-i)} \leftarrow Y^{(-i)} - M_X^{(-i)} \beta_{\theta,\lambda}^{(-i)}$
   $\tilde{q}_{\theta,\lambda}^{(-i)} \leftarrow \left\| r^{(-i)} \right\|_{K_X^{-1}}^2$
   $m_{\lambda,\theta}^{(-i)} \leftarrow K_{xX}\left(R_X^{(-i)}\right)^{-1}\left(R_X^{(-i)}\right)^{-T} r^{(-i)} + \left(\beta_{\lambda,\theta}^{(-i)}\right)^T m(x)$
   $C_{\theta,\lambda}^{(-i)} \leftarrow B(\boldsymbol{x}^{(-i)}) / [k_\theta(\boldsymbol{x}^{(-i)}, \boldsymbol{x}^{(-i)})]$
**end for**
**return** $\{m_{\theta,\lambda}^{(-i)}\}_{i=1}^n$, $\{q_{\theta,\lambda}^{(-i)}\}_{i=1}^n$, $\{C_{\theta,\lambda}^{(-i)}\}_{i=1}^n$

---

**Frozen Hyperparameters** We remark that our LOOCV algorithm is possible because sparse grids and QMC are *deterministic*—since the underlying sampling grids in hyperparameter-space are frozen—in contrast to Monte Carlo (MC) methods, which are *stochastic*. Since we use fixed sparse grids, and we are in fact interested in evaluating the posterior distribution at fixed hyper-parameters $\{\theta_i, \lambda_i\}_{i=1}^M$. If the sampling grid were not frozen across sub-models, our approach would not be viable, because the sampled points in hyperparameter-space would be different for each sub-model. Likewise, in the MLE approach, hyperparameters $\{\theta_i, \lambda_i\}_{i=1}^M$ should theoretically be retrained on the submodels, hence we cannot re-use computed values.

---

**Algorithm 2** Fast Determinant Computation

---

1: **Inputs** $K_X$
2: **Output** $\{\log |K_X^{(-i)}|\}_{i=1}^n$
3: Precompute $R^T R = K_X$
4: Precompute $\log(|K_X|)$
5: **for** $i = 1 \dots n$ **do**
6:    $b_i = e_i^T K_X^{(-1)} e_i$
7:    $\log |K_X^{(-i)}| \leftarrow \log(|K_X|) + \log(b_i)$ (Propsition B.1)
8: **end for**
9: **return**

---

## B.1 Auxiliary Results

In this section, we present linear algebra results used in the derivations of Algorithms 1 and 2 in § B.2.

**Proposition B.1** (Determinant of a Principal Minor).

$$\det\left(\Sigma^{(-i)}\right) = \det(\Sigma)\left(e_i^T \Sigma^{-1} e_i\right)$$

**Proposition B.2** (Abridged Linear System). *Let $K \in \mathbb{R}^{n \times n}$ be of full rank, and let $c, y \in \mathbb{R}^n$ satisfy $Kc = y$. Then if $r_i = c_i / e_i^T K^{-1} e_i$, we have:*

$$c^{(-i)} = (K^{(-i)})^{-1} y^{(-i)} = c - r_i K^{-1} e_i.$$

**Lemma B.1** (Determinant of the Schur Complement of a Principal Minor). *If $X \in \mathbb{R}^{n \times m}$ with $m < n$ has full column rank and $\Sigma \in \mathbb{R}^{n \times n}$ is a positive definite matrix, then*

$$\det\left(\left(X^{(-i)}\right)^T \left(\Sigma^{(-i)}\right)^{-1} X^{(-i)}\right)$$
$$= -\frac{1}{\det\left(\Sigma^{(-i)}\right)} \det\left(\begin{bmatrix} \Sigma & X \\ X^T & O \end{bmatrix}\right) e_i^T \begin{bmatrix} \Sigma & X \\ X^T & O \end{bmatrix}^{-1} e_i$$

*Proof.* Extend the Cholesky factorization $R_{11}^T R_{11}$ of $\Sigma$ to obtain the LDL-decomposition

$$W := \begin{bmatrix} \Sigma & X \\ X^T & O \end{bmatrix} = \begin{bmatrix} R_{11}^T & 0 \\ R_{12}^T & R_{22}^T \end{bmatrix} \begin{bmatrix} I & 0 \\ 0 & -I \end{bmatrix} \begin{bmatrix} R_{11} & R_{12} \\ 0 & R_{22} \end{bmatrix}$$

where $R_{22} = \text{Cholesky}(R_{12}^T R_{12})$ and $R_{12} = R_{11}^{-T} X$. Observe $\left(X^{(-i)}\right)^T \left(\Sigma^{(-i)}\right)^{-1} X^{(-i)}$ is a Schur complement of $W^{(-i)}$. This implies that

$$\det\left(W^{(-i)}\right)$$
$$= \det\left(\Sigma^{(-i)}\right) \det\left(-\left(X^{(-i)}\right)^T \left(\Sigma^{(-i)}\right)^{-1} X^{(-i)}\right)$$

By Proposition B.1

$$\det\left(W^{(-i)}\right) = \det(W) e_i^T W^{-1} e_i.$$

Therefore

$$\det\left(\left(X^{(-i)}\right)^T \left(\Sigma^{(-i)}\right)^{-1} X^{(-i)}\right)$$
$$= \frac{1}{\det(\Sigma^{(-i)})} \det(W) e_i^T W^{-1} e_i.$$

$\square$

**Lemma B.2** (Rank one downdate for bilinear forms). *If $x \in \mathbb{R}^n$ and $\Sigma$ is a positive definite matrix in $\mathbb{R}^{n \times n}$, then*

$$\left(x^{(-i)}\right)^T \left(\Sigma^{(-i)}\right)^{-1} x^{(-i)} = x^T \left(\Sigma^{-1} - \frac{\Sigma^{-1} e_i e_i^T \Sigma^{-1}}{e_i^T \Sigma^{-1} e_i}\right) x$$

*where $\Sigma^{(-i)} \in \mathbb{R}^{(n-1) \times (n-1)}$ is the $(i, i)$th principal minor of $\Sigma$ and $x^{(-i)} \in \mathbb{R}^{(n-1)}$ results from deleting the $i$th entry of $x$.*

*Proof.* By Lemma B.1, we have

$$\left(x^{(-i)}\right)^T \left(\Sigma^{(-i)}\right)^{-1} x^{(-i)} = \det\left(\left(x^{(-i)}\right)^T \left(\Sigma^{(-i)}\right)^{-1} x^{(-i)}\right)$$
$$= \frac{1}{\det(\Sigma^{(-i)})} \det(W) e_i^T W^{-1} e_i.$$

In this equation,

$$W = \begin{bmatrix} \Sigma & X \\ X^T & O \end{bmatrix} = R^T \begin{bmatrix} I & 0 \\ 0 & -1 \end{bmatrix} R, \quad R = \begin{bmatrix} R_{11} & R_{12} \\ O & R_{22} \end{bmatrix},$$

where $R_{11} = \text{chol}(\Sigma)$, $R_{12} = R_{11}^{-T} X$, and $R_{22} = \sqrt{x^T \Sigma^{-1} x}$. Using this decomposition, we may compute the term $e_i^T W^{-1} e_i$. Since,

$$R^{-T} e_i = \begin{bmatrix} R_{11}^{-T} e_i & -\dfrac{x^T \Sigma^{-1} e_i}{\sqrt{x^T \Sigma^{-1} x}} \end{bmatrix}^T,$$

we have

$$e_i^T W^{-1} e_i = e_i^T R^{-1} R^{-T} e_i - \frac{(e_i^T \Sigma^{-T} x)(x^T \Sigma^{-1} e_i)}{x^T \Sigma^{-1} x}$$
$$= e_i^T \Sigma^{-1} e_i - \frac{(x^T \Sigma^{-1} e_i)^2}{x^T \Sigma^{-1} x}.$$

Lastly, we have

$$\frac{\det(W)}{\det(\Sigma^{(-i)})} = \frac{-\det(\Sigma)\det(x^T \Sigma^{-1} x)}{\det(\Sigma) e_i^T \Sigma^{-1} e_i} = \frac{x^T \Sigma^{-1} x}{e_i^T \Sigma^{-1} e_i}.$$

These together imply that

$$\left(x^{(-i)}\right)^T \left(\Sigma^{(-i)}\right)^{-1} x^{(-i)} = x^T \Sigma^{-1} x - \frac{x^T \Sigma^{-1} e_i e_i^T \Sigma^{-1} e_i}{e_i^T \Sigma^{-1} e_i}$$

as desired. $\qquad\square$

**Proposition B.3** (Rank one matrix downdate). *If $X \in \mathbb{R}^{n \times m}$ with $m < n$ has full column rank and $\Sigma$ is a positive definite matrix in $\mathbb{R}^{n \times n}$. Let $v_i = \Sigma^{-1} e_i$ then*

$$\left(X^{(-i)}\right)^T \left(\Sigma^{(-i)}\right)^{-1} X^{(-i)} = X^T \left(\Sigma^{-1} - \frac{v_i v_i^T}{e_i^T \Sigma^{-1} e_i}\right) X,$$

*where $\Sigma^{(-i)} \in \mathbb{R}^{(n-1) \times (n-1)}$ is the $(i,i)$th principal minor of $\Sigma$ and $X^{(-i)} \in \mathbb{R}^{(n-1) \times m}$ results from deleting row $i$ from $X$.*

*Proof.* Let $\hat{\Sigma} := \Sigma^{-1} - \dfrac{\Sigma^{-1} e_i e_i^T \Sigma^{-1}}{e_i^T \Sigma^{-1} e_i}$. It suffices to prove that

$$(x^{(-i)})^T (\Sigma^{(-i)})^{-1} y^{(-i)} = x^T \hat{\Sigma} y, \quad \forall x, y \in \mathbb{R}^N$$

However, this follows from Lemma B.2, because

$$\left((x+y)^{(-i)}\right)^T \left(\Sigma^{(-i)}\right)^{-1} (x+y)^{(-i)} = (x+y)^T \hat{\Sigma}(x+y)$$

Expanding and canceling symmetric terms yields

$$\left(x^{(-i)}\right)^T \left(\Sigma^{(-i)}\right)^{-1} y^{(-i)} + \left(y^{(-i)}\right)^T \left(\Sigma^{(-i)}\right)^{-1} x^{(-i)}$$
$$= x^T \hat{\Sigma} y + y^T \hat{\Sigma} x$$

implying the result. $\qquad\square$

## B.2 Algorithm Derivation

Recall the following definitions of elements in $\{\text{TParameters}^{(-i)}\}_{i=1}^n$ from §B:

$$q_{\theta,\lambda} = \min_{\beta} \left\| g_\lambda(f_X) - M_X\beta \right\|^2_{(K_X)^{-1}} \tag{11}$$

$$\hat{\beta}_{\theta,\lambda} = \text{argmin}_\beta \left\| g_\lambda(f_X) - M_X\beta \right\|^2_{(K_X)^{-1}} \tag{12}$$

$$m_{\lambda,\theta} = K_{\boldsymbol{x}X} K_X^{-1}\big(g_\lambda(f_X) - M_X\hat{\beta}_{\lambda,\theta}\big) + \hat{\beta}_{\lambda,\theta}^T m(\boldsymbol{x}) \tag{13}$$

$$C_{\lambda,\theta} = B(\boldsymbol{x})/[k_\theta(\boldsymbol{x},\boldsymbol{x})] \text{ (Schur Complement)} \tag{14}$$

We use the generalized least squares LOOCV subroutine, outlined in Section B.2.1 to compute $q_{\theta,\lambda}^{(-i)}$ and $\hat{\beta}_{\theta,\lambda}^{(-i)}$ efficiently for all $i \in \{1,...,n\}$. We use Proposition B.2 to efficiently compute $m_{\lambda,\theta}$ and $C_{\lambda,\theta}$ whenever a perturbed linear system arises. Generally, these routines involve precomputing a Cholesky decomposition and using it for back-substitution. These steps are enumerated in Algorithm 1.

The computation of $\{\text{Det}^{(-i)}\}_{i=1}^n$ is straightforward given Proposition B.1 and a Cholesky decomposition of the kernel matrix.

### B.2.1 Generalized Least Squares

The generalized least squares (GLS) LOOCV problem is that of solving the following set of problems efficiently:

$$\left\{ \arg\min_{x \in \mathbb{R}^p} \left\| b^{(-i)} - A^{(-i)}x \right\|^2_{\Sigma^{(-i)}} \right\}_{i=1}^n.$$

It is assumed that $\Sigma \in \mathbb{R}^{n \times n}$ is positive definite, $b \in \mathbb{R}^n$, $A \in \mathbb{R}^{n \times p}$, $\text{Rank}(A) = \text{Rank}(A^{(-i)}) = p$ for some $p < n$ and for all $i \in \{1, 2, ..., n\}$.

We consider the normal equations for the $i$th subproblem:

$$\arg\min_{x \in \mathbb{R}} \left(A^{(-i)}x - b^{(-i)}\right)^T \left(\Sigma^{(-i)}\right)^{-1} \left(A^{(-i)}x - b^{(-i)}\right),$$

namely,

$$\left(A^{(-i)}\right)^T \left(\Sigma^{(-i)}\right)^{-1} A^{(-i)}x = \left(A^{(-i)}\right)^T \left(\Sigma^{(-i)}\right)^{-1} b^{(-i)}. \tag{15}$$

We first show that Equation 15 has a unique solution. By Proposition B.3, we have

$$\left(A^{(-i)}\right)^T \left(\Sigma^{(-i)}\right)^{-1} \left(A^{(-i)}\right) = A^T\Sigma^{-1}A - \frac{v_i v_i^T}{e_i^T\Sigma^{-1}e_i},$$

where $v_i = A^T\Sigma^{-1}e_i$. The LHS is a rank-1 downdate applied to $A^T\Sigma^{-1}A$. Moreover, the LHS is positive definite and hence invertible, because by assumption, $\text{Rank}(A^{(-i)}) = p$, and $\Sigma^{(-i)}$ is positive definite.

We find the solution to Equation 15 by first computing the Cholesky factorization of the LHS. Specifically, given a Cholesky factorization $R^T R$ of $A^T\Sigma^{-1}A$ from the full problem, the Cholesky factorization of the subproblem can be computed by a $\mathcal{O}(p^2)$ Cholesky downdate $R^{(-i)} = \text{Downdate}(R, v_i/e_i^T\Sigma^{-1}e_i)$ such that

$$\left(R^{(-i)}\right)^T R^{(-i)} = \left(A^{(-i)}\right)^T \left(\Sigma^{(-i)}\right)^{-1} A^{(-i)}.$$

We therefore can solve the normal equation 15

$$x^{(-i)} = \left(\left(A^{(-i)}\right)^T \left(\Sigma^{(-i)}\right)^{-1} A^{(-i)}\right)^{-1} \left(A^{(-i)}\right)^T y^{(-i)},$$

where where $y^{(-i)} = \left(\Sigma^{(-i)}\right)^{-1} b^{(-i)}$. The cost of $\mathcal{O}(n^2)$ is attained by evaluating terms from right to left. We first evaluate $y^{(-i)}$ in $\mathcal{O}(n^2)$ time by making use of Proposition B.2. We then perform back-substitution using the cholesky factor $R^{(-i)}$ in $\mathcal{O}(p^2)$ time. The overall time complexity is thus $\mathcal{O}(n^3)$.

## C   Supplementary Numerical Results

In addition to the main results in Table 4, we provide results of the MAE metrics as well in Table 6. Conclusions are similar as in Section 5.

| | IntSine | | Camel | | Abalone | | Wine | | Creep | |
|---|---|---|---|---|---|---|---|---|---|---|
| | RMSE | MAE | RMSE | MAE | RMSE | MAE | RMSE | MAE | RMSE | MAE |
| GP | 0.227 | 0.171 | 2.003 | 1.781 | 3.290 | 2.208 | 1.994 | 1.792 | 37.88 | 25.68 |
| WGP-A | 0.179 | 0.117 | 2.055 | 1.815 | 3.097 | 2.068 | 0.811 | 0.677 | 35.48 | **24.27** |
| WGP-SA | 0.172 | 0.109 | 2.012 | 1.788 | 2.992 | 2.022 | 0.809 | 0.670 | 61.62 | 45.85 |
| WGP-BC | 0.184 | 0.123 | 1.964 | 1.751 | 2.826 | 1.940 | 1.045 | 0.784 | 40.38 | 25.49 |
| CWGP-L-SA | 0.172 | 0.109 | 2.055 | 1.815 | 3.117 | 2.100 | 0.808 | 0.670 | 91.89 | 73.05 |
| CWGP-A-BC | 0.174 | 0.113 | 1.960 | 1.747 | 3.088 | 2.069 | 0.808 | 0.670 | 37.89 | 25.34 |
| BTG-I | 0.169 | 0.100 | 1.820 | 1.731 | 2.804 | 1.842 | 0.808 | 0.670 | 38.11 | 26.18 |
| BTG-A | 0.168 | 0.101 | 1.827 | 1.741 | 2.890 | 1.900 | 0.807 | 0.669 | 38.68 | 27.68 |
| BTG-SA | 0.165 | 0.104 | 1.801 | 1.796 | **2.791** | 1.822 | 0.820 | 0.696 | 91.69 | 74.04 |
| BTG-BC | 0.170 | 0.102 | 1.675 | 1.666 | 3.225 | 2.172 | 0.808 | 0.670 | 39.10 | 26.52 |
| BTG-L-SA | **0.145** | **0.082** | **1.673** | **1.658** | 2.871 | 1.870 | 0.809 | 0.670 | 91.83 | 73.05 |
| BTG-A-BC | 0.159 | 0.090 | 1.828 | 1.742 | 2.832 | **1.814** | **0.802** | **0.664** | **35.25** | 24.44 |

Table 6: A comprehensive set of RMSE and MAE results for prediction using the `IntSine`, `SixHumpCamel`, `Abalone`, `Wine` and `Creep` datasets. We generally kept the training sets small, to highlight BTG's advantages in the data-sparse regime, though for the `IntSine` function there was a good amount of data. We compare the following models: GP, WGP, CWGP, BTG, and use the following transformations transformations: I: Identity, A: ArcSinh, SA: SinhArcSinh, BC: BoxCox, L: affine. The best method is bolded; we find that BTG has lower RMSE and MAE than the other models, though unsurprisingly, different transformations are more suitable for different datasets.

