# OpenReview forum: "Bayesian Transformed Gaussian Processes"
_TMLR — Accepted by TMLR_

### Review · Reviewer_bVfk · 2023-02-22

**Summary Of Contributions:**

This paper revisits the Bayesian transformed Gaussian introduced in Oliveira et al. (1997) and presents it as a Bayesian counterpart of the Warp GP (with potentially multiple layers of transformation) by assigning a prior to the warping parameters. The main contribution of the paper is to introduce a few techniques to speed up the computational cost of BTG inference, prediction and selection, including:
- Replace Monte Carlo estimation with sparse grid quadrature or Quasi-Monte Carlo. Additionally sparsify the samples to trade-off approximation error with speed
- Use quantile bounds to speed up quantile estimation.
- Fast implementation of Leave-One-Out-Cross-Validation that reduces the computation complexity from N^4 to N^3.
The paper also conducts empirical comparison of BTG with multiple transformation layers with GP and WGP methods, and provides a Julia package for transformed GPs.

**Audience:**

Yes

**Broader Impact Concerns:**

None. This paper introduces a few techniques to speed up the computation of an existing model.

**Claims And Evidence:**

No

**Requested Changes:**

Please refer to Weaknesses section for requested change. The main points:
- Comparison with BWGP
- Experiments on additional benchmarks. For ablation studies, it’s fine to use one example for demonstration in the main text but it would be very useful to have additional examples in the appendix. For the main results in Table 4, I would expect more test functions.
- Rewriting part of the text as pointed out above.

**Strengths And Weaknesses:**

*Strengths*
- This paper is well written in general and easy to follow, although a few sections would require some extent of rewritten and clarification.
- The introduced techniques to speed up BTG computation are sound and well explained. Using quantile bounds to speed up the search is a nice idea.
- The theoretical analysis to quantify the quantile estimation error caused by sparsification is useful in practice.
- The empirical analysis of each component of the algorithm is useful for readers to appreciate their contribution to the entire method.
- The Julia package could be a useful tool for people to use more sophisticated and robust GP algorithms than a vanilla implementation.


*Weaknesses*
- Limited novelty. The main contribution is to speed up a BTG model proposed in 1997 using a series of techniques. Quadrature is a widely used method to speed Monte Carlo estimation. It’s hard to justify that revisiting BTG in the context of ML and GP literature as a contribution compared to the original paper that proposed BTG as a Bayesian version for Gaussian random fields.

- Some claims about the efficiency of the proposed method are not well supported by the experiment.
  - In Figure 3, to obtain a sufficient weight coverage, e.g. 0.9 or 0.95, the required number of nodes of QMC is similar to sparse grid quadrature. How do the authors choose the sparsification level in the experiments? Also, the prediction time is shown in the log-log scale. One should show it in the linear scale to tell whether it’s linear or estimate the exponent.
  - In Figure 4, the authors claim “It is verified that the naive way has a O(n^4) cost while our … to O(n^3)” but the figure clearly shows that Naive LOOC scales better than n^4 while the prosed is worse than n^3. Can you estimate the exponent of the curves and present them in the paper?
  - Please list the estimation error in Table 4 to check whether the differences are statistically significant.

- Lack of comparison with BWGP

The closest method to the BTG is Bayesian WGP. The authors only compare their method with vanilla GP and WGP with MLE but what they really need to compare with is BWGP that also takes a Bayesian approach to estimate the warping parameters.

- Not sufficient empirical verification.

Many of the conclusions are drawn based on a single (synthetic) example, e.g. the smoothness of the $\phi$ function in Figure 2 to justify the use of quadrature method; bounds speed up in Table 3; comparison of weight concentration in Figure 3. Also there are plenty of benchmarks for function regression, both synthetic and real ML applications. The empirical experiments on the five datasets in Table 4 (2 synthetic and 3 real) is probably not convincing enough.

- There are several places that require some rewriting.
  - The introduction of BTG in section 3 is not clear enough. A simple explanation for “trans-kriging model” would be very helpful to understand how the presentation of BTG here is different from that in Oliveira et al. (1997).
  - Section 3.1, the notation n, p, and prior of parameters in the marginal probability are used without definition. Is the notation \eta a typo? Statements like “BTG adopts the improper joint prior” and “BTG then adopts a conditional normal-inverse-gamma prior on (\beta, \tau)” are contradicting with each other.
  - “input-waped” GPs has another meaning in the literature (Snoek et al., 2014) that warps the inputs x while this paper warps the outputs y.
  - Figure 1 is not a good example to show the benefit of BTG compared to GP as “a fully Bayesian model” vs “its MLE counterpart”. To show the advantage against WGP, I’d expect a comparison with WGP instead of GP. Also, it looks like the length scale of Figure 1 is not well fitted. I guess the GP prediction will be better with a smaller length scale.
  - In section 2.3 and Table 1, “the Bayesian Warped GP is not a fully Bayesian GP, and instead approximates the posterior distribution using variational inference”. I would not say that BWGP is not fully Bayesian while the proposed method is. Neither computes the exact posterior accurately but takes a different approximation, variational inference vs Monte Carlo.
  - Conclusion: “we introduced the Bayesian transformed Gaussian (BTG) model”. I would not say so.

Refs:
Snoek, J., Swersky, K., Zemel, R. and Adams, R., 2014, June. Input warping for Bayesian optimization of non-stationary functions. In International Conference on Machine Learning (pp. 1674-1682). PMLR.

---

> ### Author Response · Authors · 2023-03-17
> **Response to Reviewer bVfk**
>
> **What about BWGP?**
>
> - We wanted to compare against BWGP, but unfortunately couldn’t find any code release (we added a footnote about this in Table 4). We did run results for DGP (Reviewer xyvo asked for them), which indicate that BTG outperforms DGP for our low-medium sized problems. We hope you will accept these as a proxy for BWGP experiments, as both BWGP and DGP have warping and use VI.
> - More generally, we do expect BWGP to perform worse than BTG in our experiments (see our note to Reviewer xyvo) for the same reasons DGP performed poorly. This is explicitly mentioned also in the CWGP paper, which didn’t bother comparing to either BWGP or DGP because of the approximation errors involved with VI.
>
> **Many of the conclusions are drawn based on a single (synthetic) example, e.g. the smoothness of the function in Figure 2 to justify the use of quadrature method; bounds speed up in Table 3; comparison of weight concentration in Figure 3.**
> - Perhaps there is a misunderstanding, Figure 2 and Figure 3 are illustrative examples, chosen to provide intuition to the reader. We tried to make this clear in the text, but will make another pass to emphasize this
> - In Table 3, you are right that we could be more thorough, but we do think the table illustrates the point we are trying to make; it’s not meant to be comprehensive, and is really just there to validate the theory we provided about the quantile bounds speed-up. If you think readers would benefit from having more comprehensive results here we will happily put some in!
>
> **Limited Novelty**
> - TLMR explicitly states that novelty is not something emphasized in the review process. So while limited novelty is a weakness in many conference venues, we don’t consider it a weakness here. We are well aware that our paper revisits an existing model— in fact, that’s why we submitted to TLMR in the first place. We have done our best to make this clear while also emphasizing the new computational techniques we developed for this model. Though they may not be very exciting, we do believe they are essential to making BTG practical, present interesting and useful theoretical results, and consequently represent a worthy contribution to the community.
>
> **Experiments on additional benchmarks… The empirical experiments on the five datasets in Table 4 (2 synthetic and 3 real) are probably not convincing enough**
> - We are not sure how many datasets you believe to be necessary in order to be convincing. We understand the importance of having a thorough empirical investigation (hence why we took the time to search for a BWGP implementation), but is an extra dataset or two really going to make the difference? We have been as thorough (if not more in our opinion) with our empirical evaluation compared to other papers such as BWGP, CWGP or DGP.
> - If the lack of additional datasets is a deal-breaker for you, then we will try our best to put in some more experimentation with the time we have remaining. But we hope you can understand our choice to address what we considered more important issues (DGP, UQ numbers, text edits) in the time that we had for a response!
>
> **Clarity/Rewrite**
> - Thank you for reading Section 3 thoroughly and for pointing out areas that need clarification. We really do appreciate it! We fixed a number of typos you mentioned, expanded our discussion of kriging and trans-kriging, modified the conclusion, and provided a more straightforward explanation of BTG priors.
> - In particular, I think the way we worded the priors was confusing. We have made this clearer in the text. The important part is that the prior on $\beta$ and $\tau$ follows a normal-inverse-gamma distribution (and it’s a conditional distribution because it depends on $\theta$ and $\lambda$). The marginal of a normal-inverse-gamma distribution is a t-distribution, hence why we get the mixture of t distributions that defines BTG’s posterior predictive density.
> - We changed the way we present “VI” and “Fully Bayesian” in the paper, as both you and Reviewer 6uRk pointed out though BWGP and DGP use VI, that doesn’t exclude them from being Bayesian. We do believe that there is a significant difference in MC vs VI, because the former is guaranteed to asymptotically converge to zero error while the latter has no such guarantees (see our note to reviewer 6uRk). At any rate, we changed our presentation because we may have appeared to dismiss these methods, when really it wasn't the case. We simply thought they were quite different.
>
> **How is sparsification level chosen in experiments?**
> - We select by the total amount of mass the selected weighted sum accounts for. For example, for a given computed weighted sum in Equation 8, we select nodes $(\theta_i, \lambda_i)$ with decreasingly ordered weights until the selected weighted sum accounts for $p$ percent (user defined) of the total sum ---$p$ is usually 98% in our experiments.

---

### Review · Reviewer_xyvo · 2023-02-28

**Summary Of Contributions:**

This paper revisited the Bayesian Transformed Gaussian process (BTG) model, which is summarized below:

The original BTG (Kedem & Oliviera, 1997) imposes a Gaussian process (GP) prior on a parametric transformation of a (non-parametric) function of interest. This helps relaxing a strong assumption of Gaussian process which stipulates that the observation distribution is a multivariate normal distribution. This is similar to the idea of warped Gaussian processes (WGP) by (Snelson et al., 2004) but unlike WGP which optimizes the transformation/warped function via MLE; the BTG model further imposes a parametric prior on the warping function, which enables analytic derivation of the model evidence.

However, the BTG model needs to use an artificial (and uninformative) prior to enable such analytic full Bayesian treatment, which is also costly in spaces of high-dimensional data. The BTG revisitation proposed in this paper therefore focuses on developing a more practical Bayesian treatment to reduce the overall computation cost of BTG while retaining its full Bayesian flavor. The BTG is also generalized to include multi-layered warping. Efficient methods for computing BTG predictive medians and quantiles as well as fast LOOCV with BTG are also proposed. The results are positively demonstrated on 6 regression datasets with input dim ranging from 5 to 18.


**Audience:**

Yes

**Claims And Evidence:**

No

**Requested Changes:**

1. Comparison with Deep GP models as well as the warping variant of sparse Gaussian processes as elaborated above.
2. Elaborate (at least empirically) on how the revisited BTG would have an advantage over deep GP as well as the (warped) sparse GP baselines in terms of uncertainty calibration.
3. If possible, do consider providing extra experiments with higher-dimensional (e.g., d = 100) datasets.

**Strengths And Weaknesses:**

Strengths:

The paper is very well-written with clear & well-organized presentation. Despite being highly technical, the paper is actually easy to follow.
The theoretical results are interesting & refreshing to read.

Weaknesses:

Despite the above strengths, there are several aspects of this paper that are unclear or even debatable for me:

First, I do not agree with the viewpoint that variational inference on deep GP is not considered a fully Bayesian treatment. To me, full Bayesian treatment means we have uncertainty calibration of the model parameters; and variational inference (VI) does provide that. In fact, exact full Bayesian treatment is rarely possible and mostly people would resort to approximation schemes which are either via VI or sampling, such as the MC approaches adopted by the authors.

Thus, in this line of thought, I would consider deep GP as a variant of multi-layer warped GP with full Bayesian treatment. As pointed out by the authors, deep GP provides a multi-layered transformation in which each layer is a random function distributed by a GP prior. On high level, deep GP and the revisited BTG basically aims to solve the same modeling problem with the same idea on provided multi-layered warping except that one is non-parametric while the other is parametric.

Given this, I believe the experiment needs comparison with deep GP baselines to demonstrate more convincingly the advantages of the revisited BTG over deep GP.

Second, regarding the improved scalability of BTG, I think the authors might have missed comparison with a set of potentially competitive baselines that involves the use of Gaussian processes. To elaborate, while warped Gaussian processes had been mostly demonstrated for full Gaussian processes, incorporating such warping idea to sparse Gaussian processes is straightforward and will likely lead to significant improvement in the compute cost.

To see this, note that the key to warped Gaussian processes is a simple application of the change-of-variable theorem to the marginal likelihood, which results in an augmentation of the original likelihood with an additional log det of the corresponding Jacobian. Most sparse GP (which scales linearly in the no. of observations) has an analytic marginal likelihood and hence, incorporating the warping idea is immediately possible. I suspect this will likely reduce the cost to O(Mn) where M is the no. of quadrature nodes.

Here are a few samples in this direction:

Sparse online warped Gaussian process for wind power probabilistic forecasting (Applied Energy, 2013, pages 410-428)
Sparse Spectrum Warped Input Measures for Nonstationary Kernel Learning (NeurIPS-20)

Alternatively, we can also pick any off-the-shelf sparse GPs [*] and apply the change-of-variable theorem on their marginal likelihood to derive fast variant of warped Gaussian processes.

[*] https://www.jmlr.org/papers/volume6/quinonero-candela05a/quinonero-candela05a.pdf

Last, the entire point of being fully Bayesian is to have good uncertainty calibration. But this aspect has not been demonstrated at all. Furthermore, the claim that the proposed method can scale well in high-dimensional data space is a not very well-demonstrated seeing that most experimental datasets have low dimensions.

---

> ### Author Response · Authors · 2023-03-17
> **Response to Reviewer xyvo**
>
> **What about DGP?**
> - We appreciate your explanation of why we ought to include DGP. We took a bit of time to run DGP (we found a nice GPyTorch implementation) on all datasets and get the following RMSE numbers: 0.241 on IntSine, 1.344 on Camel, 3.212 on Abalone, 0.824 on Wine and 91.95 on Creep. This is worse than BTG on all datasets except for Camel. This performance gap holds for other metrics and we added full results in the paper.
> - We are not too surprised by this result, as DGP uses VI. We agree that VI is an approximation method in the same way that MC is, but we do believe there’s an important difference.
> With quadrature (whether it be MC, Gauss Hermite, sparse grids, etc), we expect the approximation error to go to zero asymptotically as you increase the number of iterations. And in this paper, we have pretty good understanding (and some new theory) over the approximation errors associated with our doubly sparse quadrature; this gives us control over that error and enables us to keep it small in a principled fashion. That’s not the case with VI, which sacrifices some control for greater scalability.
> - In tougher problems, such as those with large amounts of data or high dimensionality, VI will undoubtedly prove much more useful! But certainly for the small and medium sized problems in this paper, we believe our method has showcased its strengths.
> - That being said, you and Reviewer bVfk have both requested comparisons with VI based models, and we do understand that they are a noticeable omission! We have also softened some of the text in our paper so that we do not appear dismissive of VI, as we did not mean to be.
>
> **What about incorporating sparse GPs?**
> - We are going to interpret this question more broadly as asking why we don’t focus on any scalable kernel methods, as it was also asked by Reviewer 5sB8. It’s a fair question and we will make our answer more clear in the paper; it’s only briefly mentioned in the conclusion.
> - The techniques in this paper focus on bringing down the overhead of the root finding and the integration. These are by far the dominant bottlenecks compared to a Cholesky factorization of the kernel matrix; in the case of regression, you only compute the Cholesky once. Compare this to many iterations of root finding, each of which requires integrating a mix of t distributions, for each point in your dataset, and we hope you can understand why the paper focuses on these bottlenecks!
> - Thus, we did not include any computational techniques for the kernel matrix, especially since we don’t explicitly focus on handling large amounts of data.
> - That being said, almost any scalable kernel method, whether it be a sparse approximation, inducing points, kernel interpolation, or CG, is compatible with our work, as we just use the standard Cholesky. We can certainly plug in one of these scalable methods and measure the speed-up, but I’m not sure we’ll have anything particularly interesting to say other than, “we used inducing points and now we are faster”. Furthermore, the speed-up would be marginal compared to those gained by the techniques in our paper. So we would prefer to avoid any special the treatment of the kernel system in this paper, but we’re more than happy to spend the time putting something in if you believe it to be important to readers. Please let us know!
>
> **Uncertainty quantification**
> - Reviewer 5sB8 also asked for this, and we have put in the negative log predictive density (NLPD) to measure UQ in Table 4. We believe the numbers look quite favorable!
> Dimension >= 100:
> - Unfortunately, we think 100 dimensions is too high for our methods to handle. Reviewer 5sB8 asked about the dimensionality limits of our method, and we will make this more clear. As far as our datasets are concerned, the highest dimension is 37. We could probably handle 50 dimensions, but at some point we’ll hit the curse of dimensionality and require additional assumptions to save compute time. We will make these limitations clear in the paper.

---

### Review · Reviewer_6uRk · 2023-03-05

**Summary Of Contributions:**

The article proposes a computationally efficient approximation to the Bayesian Transformed Gaussian (BTG) model (i.e. composition of a GP with a deterministic function). The proposed method approximates the Bayesian posterior by a mixture of T-distributions, leverages sparse-quadrature methods (as well as a sparsification trick) instead of naive Monte-Carlo approximations, as well as clever numerical Linear-Algebra tricks for speeding-up computations. The proposed pipeline is tested on a range of test datasets and appears to provide competitive predictive performances without sacrificing scalability.

Before gathering a few comments, I would like to point out that I am **not** an expert in GP modeling.

**Audience:**

Yes

**Broader Impact Concerns:**

NA.

**Claims And Evidence:**

Yes

**Requested Changes:**


I have a few questions: these are not necessarily requests of changes, but questions that I have asked myself while reading the manuscript -- this also indicates that some of the points below are now crystal clear in the current version of the manuscript.

1. The choice of prior distributions appears quite arbitrary (ie. chosen to ease computation, mainly). Is this correct? Does it impact the performance? If scalability were not an issue, are there better choices to consider?
2. In Equation (7) for $\phi(f|\theta, \lambda, \ldots)$ , am I correct assuming that the integral with respect to $f(x)$ does not equal one, because of the term $|g'_{\lambda}(f(x))|$ ?
3. In Equation $(8)$ do the weights $\widetilde{w}_i$ sum-up to one? I am finding this confusing.
4. Why use both MSE and RMSE?
5. I am not an expert in GP -- I could not help myself wonder whether choosing a more flexible family of kernel function would be enough to avoid using any "wrapping" function? In other words, would this lead to roughly equivalent predictive performances? All the experiments are performed with the extremely simple RBF kernel.
6. Performing a simple Gaussian approximation (eg. Laplace approx) of the posterior is a possible approach in between MLE and the proposed approach -- how is this approach performing, numerically?
7. One a possibly very simple example, this would be interesting, I think, to investigate WGP vs BTG as the number of points increases -- this would help to understand the regime when the "full Bayesian treatment" starts to not be necessary.

I have enjoyed reading the text and I am looking forward to reading the authors' comments and clarifications.


**Strengths And Weaknesses:**

I have found the text well-written and relatively easy to follow. The motivation for the proposed methodology is clear, and the brief literature review is sufficient. I have not checked every single computation line-by-line, but the derivations are relatively standard. The numerical experiments are well executed and convincing. I think that it would be interesting to understand in more details the regime when the Bayesian treatment (BTG) is actually worth implementing when compared to the point-estimate approach (WGP/MLE). It is clear from the simulation (and the discussion in the text) that the data-scarce regime is such a scenarios -- what does happen with larger datasets?

---

> ### Author Response · Authors · 2023-03-17
> **Response to Reviewer 6uRk**
>
> **What happens with larger datasets?**
> - We expect a sufficiently large dataset to lead to a more tightly concentrated likelihood. Assuming this occurs, the difference between integrating over the posterior (fully Bayesian) and taking the maximal point (MLE) will not be as pronounced. This in turn will cause the performance gap between BTG and WGP to shrink. More generally, for a sufficiently large dataset we expect BTG and WGP/CWGP to look quite similar (in the same way that this occurs between a fully Bayesian GP and an MLE GP once there is enough data).
>
> **Choice of prior, Eq 7 and Eq 8**
> - If you are referring to the normal-inverse-gamma prior on (beta, tau), then yes, that prior is chosen specifically! We mention this a bit in the text at the top of page 7, but the marginal of this prior is our t distribution in equation 7. And the t distribution itself ought to integrate to one, but you’re right, the transformed t distribution in equation 7 ought to be improper
>
> **Do weights in equation 8 sum to one?**
> - The weights should not sum to one. Sorry for the confusion, there are two weights, the first is the regular w_i, which are the quadrature weights and if you do something like Monte Carlo, they should sum to one. But we combine the quadrature weights and all other values into a more general weight \tilde w_i, and these should not sum to one
>
> **Both MSE and RMSE – just for reference, because there could be different things they measure?**
> - We included both just to be thorough. At other reviewers’ requests, we have also added the negative log predictive density (NLPD) to evaluate uncertainty quantification. These NLPD numbers have replaced the MAE numbers in Table 4 (the MAE has been moved to the appendix for space’s sake).
>
> **Would a more complex kernel be able to replace the need for a warping function?**
> - This isn’t the most comprehensive answer, but in general we would expect the kernel to measure correlation between x values, whereas the warping function would give you some degree of control over the y values. So they are at first glance, unrelated. That being said, it’s an interesting question of whether or not a poor kernel choice + warping would compare against a proper kernel choice + no warping. Our intuition says it would really depend on the application and you could construct adversarial examples for each.
>
> **Performing a simple Gaussian approximation (eg. Laplace approx) of the posterior is a possible approach in between MLE and the proposed approach -- how is this approach performing, numerically?**
> - It’s a good question and one we haven’t thought of. In theory it could be quite poor, especially because the posterior of BTG could have arbitrarily high skew, but we have no intuition for how well it may work in practical situations.
>
> **Increase the number of points to see when "full Bayesian treatment" starts to not be necessary.**
> - That is an interesting experiment we think the readers will appreciate, and if we have remaining time we’ll try to put something in!
> - Generally, we observed in our numerical experiments that with more than 200 training points, the performance gap between the fully Bayesian and MLE approaches tended to shrink. We doubt this is a general trend (more a product of the small and medium sized problems we chose), as it likely depends greatly on the problem at hand; a very simple regression task might not see any improvement from a fully Bayesian treatment at all.

---

### Review · Reviewer_5sB8 · 2023-03-11

**Summary Of Contributions:**

This work revisits the BTG model, which has latent and observations spaces with a deterministic function mapping these two spaces. The central idea is to approximate the BTG posterior predictive density using a mixture of t-distributions. In doing so, several tricks are cleverly intermixed to address computational challenges in obtaining the GP posterior. The proposed work is evaluated in various relevant experimental settings to ascertain the advantages of introduced techniques, i.e., sparse grids, QMC, and fast way to perform LOOCV.

**Audience:**

Yes

**Broader Impact Concerns:**

I don't see it as a concern.

**Claims And Evidence:**

No

**Requested Changes:**

One of the most critical changes is demonstrating that the proposed approach can indeed yield better uncertainty estimates on a real dataset.

For the remaining changes, please refer to the Weaknesses section for the requested change. Consider points with which you agree.

Minor typo: 'quartic' in "which incurs quartic cost naively."


**Strengths And Weaknesses:**

Strengths:
- This paper presents relevant literature and content in a way that is easier to follow and sets up a clear motivation.
- This work presents several theoretical and practical results that are interesting. E.g., analysis to quantify the quantile estimation error by sparsifying quadrature rules is interesting.
- Empirical analyses highlighting proposed algorithms' computational and speed aspects are convincing.

Weaknesses:
- Time and accuracy compared with the MLE approach would be helpful. A top example demonstrates an 'overconfidence problem with the MLE approach, but an experimental setup involving a real dataset would add more value.
- It seems worth mentioning about GraBhoff et al. 2020 and how the author's work compares to theirs. GraBhoff et al. 2020 adopt SKI to WGP setting and is fast low dimensional settings similar to the proposed work and sparse grids.
- Simone et al. also propose a fully Bayesian sparse GPs approach that is relatively more scalable. It is worth mentioning why this approach would not apply to wrapped input settings considered in this paper. Or if it is, then it may be compared with them.
- It'd be nice to mention when these approaches are more feasible and when they won't be possible. E.g., sparse grids are typically infeasible for high dimensions. Does the sparsification of quadrature allow it to go to higher dimensions?

GraBhoff et al. 2020:- Jan GraBhoff, Alexandra Jankowski, Philipp Rostalski, "Scalable {G}aussian Process Separation for Kernels with a Non-Stationary Phase" Proceedings of the 37th International Conference on Machine Learning, PMLR 119:3722-3731, 2020.

Rossi, Simone, et al. "Sparse Gaussian processes revisited: Bayesian approaches to inducing-variable approximations." International Conference on Artificial Intelligence and Statistics. PMLR, 2021.

---

> ### Author Response · Authors · 2023-03-17
> **Response to Reviewer 5sB8**
>
> **Uncertainty estimates**
> - We agree that uncertainty quantification is one advantage of being fully Bayesian that we haven’t touched upon in this paper. We appreciate the fact that you and another reviewer pointed this out, and we believe including a discussion on this will greatly benefit the reader.
> - We took some time to compute the negative log predictive density (NLPD) to evaluate the UQ. These NLPD numbers have replaced the MAE numbers in Table 4 (the MAE has been moved to the appendix for space’s sake). The results confirm that BTG (and more generally, being fully Bayesian) tends to have better UQ; this is most pronounced for the Camel dataset, in which BTG achieves an NLPD of 0.659, whereas most of the other models have NLPD values ranges from 13-45. More generally, BTG has the best NLPD for all but one dataset, and for the last dataset the performance gap seems quite small. Please see the updated paper submission for full results.
> - We have updated the text of the paper corresponding with the analysis of Table 4. We also added a brief discussion about UQ in a few other parts of the text
>
> **Time and accuracy compared with the MLE approach would be helpful**
> - We do have end-to-end timing tests in Table 5, which indicate that BTG is competitive with competitors like WGP and CWGP, is that what you were looking for?
>
> **A toy example demonstrates an 'overconfidence problem with the MLE approach, but an experimental setup involving a real dataset would add more value.**
> - That is true, but it is simpler to carefully construct a low-dimensional toy example that convincingly illustrates our point and is easy to visualize (preferably 1d), rather than examining real-world datasets until we find a winner. We hope our more comprehensive empirical studies later on convince the reader of the overconfidence problems associated with MLE!
>
> **GraBhoff et al. 2020**
> - The SKI method is used to make the model scalable for large training data, which isn’t the focus in this paper. Additionally, SKI or any other scalable kernel method is rather orthogonal to this work. We focus on the bottleneck of integration + root finding, which created far more difficulties than those we encountered with the kernel system. See our longer note to Reviewer xyvo who asked a similar question about sparse GPs. The TL;DR is that we are happy to put in a more interesting treatment of the kernel system if you think it’s important, though we would prefer avoiding this if possible since it doesn’t relate to this paper’s primary contributions (though it is important follow-up work!).  Plus, SKI specifically wouldn’t be suitable because kernel interpolation struggles in dimensions greater than 4 or 5; something like a Nystrom or inducing point approximation would be more natural.
>
> **Simone et al. 2021**
> - We added this as a related work for discussion. It uses a fully Bayesian treatment of both the inducing points and model hyperparameters, based on stochastic gradient Hamiltonian Monte Carlo. Theoretically, this treatment should be applicable to combine with warping functions as well, but the inclusion of the warping parameters would complicate the likelihood and its gradient; such an extension would require a separate research investigation beyond the scope of this submission. More practically, similar to GraBhoff et al., this paper’s goal seems to be scaling to large datasets.
>
> **It'd be nice to mention when these approaches are more feasible and when they won't be possible. E.g., sparse grids are typically infeasible for high dimensions. Does the sparsification of quadrature allow it to go to higher dimensions?**
> - The dataset with the highest dimension in this paper is 37 (the creep dataset with composed transformation “Affine+BoxCox”). You’re right to note that sparse grids stop being effective past a certain point. The sparsification scheme we developed can push past this limit because we do not have to repeatedly compute integrals as a huge sum, but it too has a limit (perhaps no more than 50 dimensions, as 37 was challenging enough). We made this more clear in the paper.
> - We will also reinforce that we aim for the low-medium data size/dimension regime in which we know we can provide concrete benefits over existing models and methods. We mentioned in the text already that we want to be transparent about the strengths and weaknesses of our paper, and the high-d regime is certainly one situation we struggle with.
> - That being said, due to the curse of dimensionality, it is hard to do sparse grid quadrature, or any other quadrature rule in high-d without additional assumptions such as the shape of the integrand. Even whether such assumptions exist for general problem classes is unclear for regular fully Bayesian GPs, to say nothing of BTG!

---

### Decision · Action_Editors · 2023-04-04

**Recommendation:** Accept as is

**Comment:**

This submission makes an important contribution to a remarkably flexible model class: the Bayesian transformed Gaussian (BTG) model. Although quite flexible and useful for modeling, the model presents numerous computational challenges in a practical setting. The authors outline best practices for working with this model -- introducing some useful computational "tricks" along the way -- resulting in an effective scheme for working with the BTG model in practice.

These algorithmic contributions are coupled with a series of well-designed empirical investigations offering additional insight into the BTG model.

I want to commend the authors in particular for their efforts in engaging the reviewers during the discussion period and incorporating numerous edits into their draft accordingly, which I believe strengthened the paper considerably.

**Audience:**

There is no question that the material in this paper would be of interest to a significant subset of TMLR's audience, as it concerns fundamental techniques for Bayesian modeling.

**Claims And Evidence:**

The claims made in this submission are supported both by clear and convincing arguments and well-designed and insightful empirical studies.